# Mutation Associated with Orange Fruit Color Increases Concentrations of β-Carotene in a Sweet Pepper Variety (*Capsicum* *annuum* L.)

**DOI:** 10.3390/foods10061225

**Published:** 2021-05-28

**Authors:** Nasya Tomlekova, Velichka Spasova-Apostolova, Ivelin Pantchev, Fatma Sarsu

**Affiliations:** 1Department of Breeding, Variety Maintenance and Introduction, Maritsa Vegetable Crops Research Institute, 32 Brezovsko Shosse St., 4003 Plovdiv, Bulgaria; 2Department of Breeding and Seed Production, Tobacco and Tobacco Products Institute, 4108 Markovo, Bulgaria; vilispasova-apostolova@abv.bg; 3Department of Biochemistry, Sofia University, 8 Dragan Tzankov, 1164 Sofia, Bulgaria; ipantchev@gmail.com; 4Plant Breeding and Genetics Section, Joint FAO/IAEA Division of Nuclear Techniques in Food and Agriculture, International Atomic Energy Agency, 1400 Vienna, Austria; F.Sarsu@iaea.org

**Keywords:** carotenoids, β-carotene hydroxylase, sweet pepper, mutagenesis, high performance liquid chromatography, molecular characterization

## Abstract

Pepper is the second most important vegetable crop in Bulgarian agriculture and has become the subject of extensive breeding programs that frequently employ induced mutagenesis. The success of breeding programs can be enhanced by the efficient and integral application of different biochemical and molecular methods to characterize specific mutant alleles. On the other hand, identifying new cost-effective methods is important under a limited-resources environment. In this paper we compare the levels of five health-related carotenoid compounds of fruits (α-carotene, β-carotene, lutein, β-cryptoxanthin, zeaxanthin) between a mutant variety Oranzheva kapia (possessing high ß-carotene concentration) and a corresponding initial pepper variety Pazardzhishka kapia 794. Both varieties are intended for fresh consumption. Pepper is a major natural source of β-carotene. It was observed that fruit at both commercial and botanical maturity from mutant variety had greater α-carotene and β-carotene concentrations to the initial variety (7.49 and 1.94 times higher, respectively) meaning that the mutant was superior in fruit quality to the initial genotype. Two hydroxylase enzymes, converting α- and β-carotene to lutein and zeaxanthin, respectively, are known to exist in pepper and are encoded by two genes on chromosomes 3 and 6-*CrtZ*_chr03_ and *CrtZ*_chr06_. The molecular characterization of the mutant variety through locus-specific Polymerase chain reaction amplification, gene cloning and sequencing as well as expression was performed. Our results suggest that the increased ß-carotene accumulation in the mutant variety Oranzheva kapia results from a biosynthetic pathway breakdown due to deletion of *CrtZ*_chr03_ gene.

## 1. Introduction

Peppers (*Capsicum* spp.) are a good source of dietary carotenoids, which has led to extensive research on their carotenoid biochemistry [1]. Since their initial domestication in Mexico, peppers have been under strong selection for shape, size, and, recently, nutritional value [2]. While wild relatives and landrace peppers are frequently small and highly pungent, bred peppers have an endless array of phenotypic diversity [3]. Efforts have been made to link genetic data to morphology using computer analysis, the first step toward a pepper phenome [4].

The economic importance of peppers is increasing, and the production is expanding. Until recently, the breeding programs were based mainly on using natural sources of germplasm, cross-breeding and the heterosis effect of F_1_ hybrids. Based on the recent successes there is growing interest in taking advantage of induced mutations [5]. Breeding programs have begun using complementary genomic approaches and efficient phenotypic and biochemical screening methods to identify and develop mutants with traits of interest such as vitamins and minerals [5]. The function of individual genes, such as those related to carotenoid pathways, can be investigated either through biochemical studies or gene introgression into other model organisms such as tomato, tobacco, Arabidopsis or bacteria [6,7,8]. The biochemical and genetic mechanisms of carotenoid biosynthesis in tomatoes are relatively well studied [9], but for peppers it is still difficult to establish links between these genes and their phenotypic expressions. This is mostly due to the absence of appropriate mutants and the inefficiency of the current transformation techniques. Hence, the development and availability of such specific mutants would be of great significance for the advancing pepper genetics and biotechnology. Within the past few years, a number of agronomically useful mutants have been induced and characterized, in addition to mutants that are valuable for genetic, cytological, biochemical, molecular and physiological studies [5].

The different colors of mature pepper fruits—red, salmon-red, pink, orange, lemon yellow, white or dark-brown—are of commercial importance and are determined by the composition of carotenoids [10]. Carotenoids are integral and essential components of the photosynthetic membranes in all plants and have a complex biosynthetic grid [11]. Each branch in this pathway offers a chemical step amenable to combinatorial biosynthesis [12].

One of the significant challenges in breeding peppers for improving quality is the concentration of β-carotene in the fruits [13]. The richest natural source of β-carotene is the paprika for grinding. The small quantities of paprika used as a spice do not meet the β-carotene needs in human food. Since pepper for fresh consumption is consumed in big quantities, it is the main source, which provides β-carotene in the human diet [14]. Thus, genotypes bred for fresh consumption should have certain levels of β-carotene (provitamin A).

Increasing carotene has always been one of the most important traits in Bulgarian pepper breeding programs [13]. X-rays were applied to the initial Pasardzhishka kapia 794 pepper variety during the 1980s. Later, the orange-fruit variety Oranzheva kapia was developed that had more β-carotene [14]. Early investigations suggested an inactivation of the carotene biosynthetic pathway during the β-carotene hydroxylation stage. The fruit color is determined by capsantin/capsorubin synthase (*Ccs*), phytoene synthase (*Psy*), lycopene-β-cyclase (*Lcyb*) and β-carotene hydroxylase (*Crt*Z) [15]. In peppers, two genes for encoding β-carotene hydroxylase have been identified on chromosomes 3 and 6 [16]. Changes in gene activities can lead to changes in carotenoid concentration [17], subsequent to color variations. On the other hand, the orange color may also be due to the mutation within either the *Psy* coding sequence or regulatory regions [18,19]. A more detailed characterization of the mutant allele is required.

The mutant variety Oranzheva kapia used in this experiment was developed by application of 120 Gy X-ray to pepper seeds of local variety Pazardzhishka kapia 794, a commonly used local variety in Bulgaria. Both belong to the group of sweet peppers ‘kapia’ type intended for fresh consumption. Development of the altered genotype was achieved [20,21,22]. M_1_ generation was observed for dominant mutations (rare cases). In M_2_ generation, recessive mutations were recorded and selection performed, and the orange fruit color was observed and selected. Altered genotypes in M_3_ were confirmed. The populations raised from M_3_ seeds were screened for various parameters for a selection of valuable characteristics following a descriptor list derived from the IPBGR (1996) for *Capsicum* [23]. The mutant plants were back-crossed with initial (three generations) and self-pollinated for a total of eight generations (M_8_), thus generating NIL/RIL [20]. Selection was directed at retaining the orange color. Developed as NIL/RIL, the mutant variety Oranzheva kapia was genetically very close to the initial Pazardzhishka kapia 794, except the orange fruit color.

Kurtovska kapia 1 is a commonly grown local variety that has been developed by the method of continuous individual selection and maintained in the Maritsa Vegetable Crops Research Institute. It also belongs to the ‘kapia’ type, a sweet pepper.

The goal of this research was to evaluate carotenoid levels biochemically in the initial variety Pazardzhishka kapia 794 and mutant variety Oranzheva kapia sweet pepper genotypes. This reveals the genetic changes that result from induced mutagenesis, which boosts the carotene health-related compounds in fruit. In the second stage of this study we present current progress in molecular analysis of the mutant variety with high β-carotene concentration.

## 2. Materials and Methods

### 2.1. Plant Material

The research was carried out on *Capsicum annuum* Linn. sweet pepper ‘kapia’ type that is commercially destined for fresh consumption. Initial and mutant genotypes included in this study- were Pazardzhishka kapia 794 and Oranzheva kapia, respectively. To assess target gene sequences, we also compared the local variety Kurtovska kapia 1.

Growth conditions and harvested parts of plants. 

Ten plants of each genotype were grown in the field at the Maritsa Vegetable Crops Research Institute, Plovdiv, Bulgaria. The plants were grown two successive years in furrows following conventional practice for mid-early pepper production in this region. 

For carotenoid analysis, twenty fruits were collected for each variety at both botanical maturity and commercial maturity stages. Fruits were selected to be of uniform size and color (Figure 1). At the same time points, ten young leaves were also collected from each variety (one per plant). All fruit and leaf samples from mutant and initial genotypes were individually lyophilized and stored in the dark in covered aluminum containers at 4 °C (Figure 2).

### 2.2. Biochemical Evaluation of Certain Carotenoid Levels through a High-Performance Liquid Chromatography (HPLC) Method

In this study, high performance liquid chromatography (HPLC) was used to evaluate the biochemical level of some carotenoids [24,25].

#### 2.2.1. Chemicals

Standards of lutein, zeaxanthin, β-cryptoxanthin, lycopene, β-carotene and α-carotene were acquired from CaroteNature (GmbH, Lupsingen, Switzerland). HPLC-grade solvents including acetonitrile, methanol and ethyl acetate were obtained from Merck (KGaA, Darmstadt, Germany).

#### 2.2.2. Sample Extraction

Carotenoids were extracted from 0.1 g ground freeze-dried samples in duplicate, to which 0.2 g magnesium carbonate had been added, with 15 mL extraction solvent methanol: tetrahydrofuran (1:1, *v*/*v*) according to Hart and Scott [26], containing 0.1% butyl hydroxytoluene (BHT) by homogenizing for 3 min at 2500 rpm in a Vortex homogenizer, 3 min at 3000 rpm centrifuged, and the supernatant was collected. The extraction was repeated until the extract was colorless. Each fruit was treated separately.

#### 2.2.3. Saponification

Extract from the extraction procedure was saponified for 1 h with an equal volume of 10% methanolic potassium hydroxide, containing 0.1% butylhydroxytoluene, in the dark at room temperature. Carotenoids were extracted twice from the potassium hydroxide/methanolic phase with diethyl ether, containing 0.1% BHT and 10% sodium chloride solution in a separating funnel. The combined diethyl ether phase was washed several times with water until neutral pH, dried over anhydrous sodium sulfate and concentrated to dryness in a rotary evaporator at 35 °C. The residue was dissolved in acetonitrile-methanol-ethyl acetate (60:20:20, *v*/*v*/*v*). The sample solution was filtered through a 0.45 Nylon membrane filter (Millipore), and 20 μL was injected for HPLC analysis.

#### 2.2.4. Chromatographic Separation

HPLC analysis was performed on a Hewlett Packard separation module (model 1050, Agilent Technologies, Santa Clara, CA, USA), equipped with a quaternary pump system, a UV–vis detector (Hewlett-Packard, series 1050) and a column heater (Hewlett-Packard, series 1100 with a Hypersil C18, 5 μm, Palo Alto, CA, USA) (250 × 4.6 mm id., Merck, Kenilworth, NJ, USA) column. Data analyses were carried out using HP ChemStation software (Agilent Technologies, Santa Clara, CA, USA).

The separation protocol was adapted from [11]. The mobile phase consisted of acetonitrile, methanol, ethyl acetate of acetonitrile-methanol (95:5, *v*/*v*) (eluent A) and acetonitrile-methanol-ethyl acetate (60:20:20, *v*/*v*/*v*) (eluent B), containing 0.1% BHT as an antioxidant and 0.05% triethylamine (TEA). A concave gradient from 100% A to 100% B was applied for 14 min until the end of the run. Separation was performed at a flow rate of 1 mL/min. The column temperature was maintained at 20 °C during the separation process. Injection volume was 20 μL. Elution profile was recorded by a UV–vis detection module set at the wavelength of 450 nm.

The average short-term reproducibility (CV%) was 6.0%. The average long-term reproducibility was 7.4%. The recovery carotenoid rates were 95% for lutein, 96% for zeaxanthin, 97% for lycopene, 102% for β-cryptoxanthin, 104% for α-carotene and 95% for β-carotene. Recovery rates were determined by spiking samples with several concentrations of each carotenoid (0.07 to 0.5 μg/mL).

Limits of detection (LOD) were estimated using a signal-to-noise ratio of 3:2 ng for lutein, 2 ng for zeaxanthin, 3.5 ng for lycopene, 2 ng for β-cryptoxanthin, 2 ng for α-carotene and 2.5 ng for β-carotene.

#### 2.2.5. Preparation of Standard Carotenoid Solutions and Calibration

Stock solutions of individual pigments (lutein, zeaxanthin, β-cryptoxanthin, lycopene, β-carotene and α-carotene) were prepared by dissolving 1 mg of each in 10 mL chloroform, containing 0.1% BHT. This solution was prepared monthly.

A working standard solution was prepared daily from individual stock solutions by evaporating aliquots with nitrogen and diluting with acetonitrile:methanol:ethyl acetate (60:20:20, *v*/*v*/*v*). Several dilutions were prepared with different concentrations of the carotenoid standards for 0.03 to 5 μg/mL.

#### 2.2.6. Analysis of the Variation

The average mean and the significance interval of the studied characteristics were calculated by descriptive statistics (SigmaPlot, SPSS Inc. and IBM, Chicago, IL, USA) released 2008). Carotenoid levels were evaluated in individual and group samples [27].

### 2.3. Molecular Characterization of β-Carotene Hydroxylases

#### 2.3.1. DNA and RNA Isolation

Genomic DNA was isolated from freeze-dried young leaves according to [28,29]. Lambda DNA (Thermo Fisher Scientific, Lithuania) was used to determine DNA quantity on 1% LE agarose gels (Lonza, Walkersville, MD, USA) with ethidium bromide (VWR International, Vienna, Austria). DNA concentration in the samples was estimated by comparing band intensity of probes with lambda DNA of known concentrations (50, 20, 10 and 5 ng/μL). Three microliters of each dilution was loaded per lane.

Fresh samples were frozen in liquid nitrogen and stored at −80 °C. Total RNA was isolated from the pericarp after grinding in liquid nitrogen. RNA from a 50 mg sample was purified with a Qiagen RNase Plant Mini Kit. RNA concentration was determined spectrophotometrically at 260 nm. Isolated RNA samples were stored at −80 °C.

#### 2.3.2. Primers

Gene-specific primers for β-carotene hydroxylase genes were located on chromosomes 3 and 6 of the pepper genome (Appendix A).

The gene for Elongation Factor-1 alpha (EF1-At1g07940) was used as a RT-PCR control. EF-1 was amplified using primers EF1-FW: ATTGTGGTCATTGGYCAYGT and EF1-RE: CCAATCTTGTAVACATCCTG [30,31].

#### 2.3.3. Gene Cloning and Sequencing

Primers used for cloning and sequencing of the *Crt*Z_chr03_ gene are shown in the “Appendix A”.

Clones derived from the variety Oranzheva kapia were analyzed by primer sets CrtZ–D_F/b-CRT 8, CrtZ-E/b-CRT 4 and b-CRT 7/CrtZ-C_R in the mutant genotype of Oranzheva kapia.

Clones derived from Pazardzhishka kapia 794 were analyzed by primer sets b-CRT 1/b-CRT 4 and CrtZ–D_F/b-CRT 8.

Primers FW1_H2/b-CRT 8 were used for *Crt*Z_chr03_ cloning and sequencing on all three plant genotypes (Pazardzhishka kapia 794, Oranzheva kapia and Kurtovska kapia 1).

Primers FW1_H1/RE1_H1 and FW1_H1/RE2_H1 were used for *Crt*Z*_chr06_* cloning and sequencing on all three plant sources (Pazardzhishka kapia 794, Oranzheva kapia and Kurtovska kapia 1).

Primer FW1_H2 was designed according to the published sequence with NCBI accession number Y09225 [32] and accession numbers GU122940, GU122941, GU122942, GU122943, GU122944, GU122945 and GU122946 [33].

Primers FW1_H1, RE1_H1 and RE2_H1 (Appendix A) for cloning and sequencing *CrtZ*_chr06_ were designed according to NCBI accession number Y09722 [32] and the Pepper Genome Database sequence of the cultivated pepper Zunla-1 (*C. annuum* L.) http://peppersequence.genomics.cn/page/species/index.jsp (last accessed on 10 June 2014) [34].

#### 2.3.4. Gene-Specific PCR

PCR reaction was performed in 25 μL volume. DNA template was 50 to 100 ng per reaction. Amplifications were performed under the following conditions: 3 min of initial denaturation at 96 °C; 35 cycles consisting of 30 s of denaturation, 30 s annealing and 1 to 4 min of elongation at 72 °C; followed by final elongation at 72 °C for 5 min. Annealing temperatures varied according to primer Tm. Elongation time varied between 1 min and 4 min depending on product length. Twenty-five PCR reactions were performed with primer combinations in order to produce overlapping fragments.

#### 2.3.5. RT-PCR and cDNA Synthesis

RT-PCR was performed on 300 ng total RNA using the One-step RT-PCR Kit (Qiagen, Valencia, CA, USA). To analyze *CrtZ*_chr03_ expression, the primer pair CrtZ-D_F/CrtZ-D_R was used. To analyze CrtZ_chr06_ expression, the primer pair BCH1F/ BCH1R was used (Appendix A).

*CrtZ*_chr03_ and *CrtZ*_chr06_ cDNA regions were amplified from total RNA using the One-step RT-PCR kit (Qiagen, Valencia, CA, USA) according to the manufacturer instructions. Briefly, about 1 μg total RNA was mixed with 0.4 μM primers (CrtZ-D_F/CrtZ-D_R for *CrtZ*_chr03_ gene or BCH1-F/BCH1-R for *CrtZ*_chr06_ gene). The mix was incubated for 15 min at 95 °C and cooled on ice. Then, 5 μL One-step RT-PCR buffer, 1 μL 10 mM dNTP and 1 μL One-step RT-PCR enzyme mix were added along with ddH_2_0 to a final volume of 25 μL.

Reverse transcription was performed for 25 min at 20 °C and then denatured for 5 min at 94 °C. Then, 35 amplification cycles were performed as follows: 1 min at 94 °C, 50 s at 55 °C, 2 min at 72 °C, followed by final extension for 7 min at 72 °C.

The Elongation factor 1 alpha (EF1-At1g07940) was used as positive control [30,31] following the same procedure.

#### 2.3.6. Gene Cloning

Amplified fragments were cloned into pTZ57R plasmid vector using InsTAclone kit (Thermo Fisher Scientific, Vilnius, Lithuania) and transformed in *Escherichia coli* DH5alpha. Plasmids from positive clones were isolated (QIAprep Spin Miniprep Kit, QIAGEN) and sent for sequencing (StarSeq GmbH, Mainz, Germany).

#### 2.3.7. DNA Electrophoresis, Ethidium Bromide Staining and Gel Documentation

Amplification products from gene-specific and RT-PCR amplifications were separated on 1.5% agarose gel (1x TAE) containing 0.5 μg/mL ethidium bromide. Size was compared with a DNA Ladder Gene Ruler 100 bp Plus DNA and GeneRuler 1 kb DNA Ladder, ready to use for RT-PCR (Thermo Fisher Scientific, Vilnius, Lithuania). Images were captured with the Azure Biosystem 600 (Biosystems, Dublin, OH, USA).

## 3. Results

All local varieties, including Pazardzhihka kapia 794 (initial variety in the mutagenesis program), had mature red fruit, while variety Oranzheva kapia (mutant variety) had orange fruit (Figure 1).

### 3.1. Biochemical Evaluation of Certain Carotenoid Levels through a High-Performance Liquid Chromatography

The pigment concentration in the mutant was analyzed by HPLC-DAD. Carotenoid standards were selected to cover anticipated pigment changes, affected by mutation. The response of lutein, zeaxanthin, β-cryptoxanthin, lycopene, α-carotene and β-carotene was linear for 0.03–5 μg/mL (r^2^ > 0.999).

Limits of quantification (LOQ) were set at 2.5 times the LOD, which were 0.025 g/mL for lutein, 0.0225 g/mL for zeaxanthin, 0.043 g/mL for lycopene, 0.024 μg/mL for β-cryptoxanthin, 0.027 g/mL for α-carotene and 0.029 μg/mL for β-carotene.

HPLC results showed that all green fruits (at botanical maturity) contained mostly β-carotene and lutein (Table 1). α-carotene, β-crypthoxanthin and zeaxanthin were not detected at this stage of fruit development (green maturity fruits).

According to the HPLC results, β-carotene and lutein were present in the green fruits of both varieties (Table 1). The mutant variety had more β-carotene and lower lutein than the initial one. In the mutant Oranzheva kapia the β-carotene was 2.25 times higher and the lutein was 1.65 times lower compared to Pazardzhishka kapia 794.

The evolution profile of carotenoids through ripening to botanical maturity stage was most probably different among the samples of initial and mutant varieties. Our findings show that the amount of some components from the keto-carotenoid group increased significantly during the early ripening phases (Table 1). The ripening of the mutant fruits is connected with a decrease in the lutein and a decrease in the relative concentrations of other hydroxyxanthophylls quantified in the study. During the ripening, the concentrations of β-carotene and the ketocarotenoids increased significantly. A plausible explanation is that the mutation in Oranzheva kapia variety had affected only the gene responsible for the hydroxylation of β-carotene to β-cryptoxanthin (Table 2).

Additionally, α-carotene levels increased in the mutant. In the initial variety, concentrations were very low (0.090 mg/100 g dry mater), while in the mutant this figure reached 0.674 mg/100 g dry mater (Table 2).

Biochemical analysis of botanically ripe fruits demonstrated the presence of α-carotene, β-carotene, β-cryptoxanthin, zeaxanthin and lutein (Table 2, Figure 3). We also found the lycopene concentration was below LOD (0.002 μg for lycopene), and it could not be detected under the analytical conditions of this study. The β-carotene concentration significantly differed between both varieties (from 3.761 mg/100 g dry matter in the initial variety to 7.309 mg/100 g dry matter in the mutant one, which was 1.94 times higher). Increased β-carotene concentration in the mutant corresponded with a decrease in β-cryptoxanthin and zeaxanthin levels. Beta-cryptoxanthin was not detected in the mutant. When Pazardzhiska kapia 794 and Oranzheva kapia were compared, the intervals of average values of the quantified carotenoid substances did not have a common point and did not intersect. The obtained data from the confidence intervals of the average values proved that the differences between initial and mutant varieties were statistically significant.

### 3.2. Molecular Characterization of the Mutation, Leading to Orange-Colored Fruits

#### 3.2.1. PCR amplifications of *CrtZ*_chr03_ and *CrtZ*_chr06_ gene

The attempts to clone cDNA from mutant *Crt*Z_chr03_ β-carotene hydroxylase gene were unsuccessful, as were subsequent attempts to amplify exons directly from gDNA. In both cases, amplifications from initial and control plants were successful. We have designed a set of primers according to the published genomic DNA sequences covering not only the coding region but also fragments from chromosome 3,20 kbp up- and downstream from *Crt*Z_chr03_ gene [35]. Amplification of ten loci within *Crt*Z_ch03_ had produced unexpected patterns, some of them with low intensity on the gel for visualization of the PCR products (Figure 4).

In the initial genotype, Pazardzhishka kapia 794, bands with expected lengths were produced at all analyzed loci. In contrast, amplification of the same loci in the mutant genotype Oranzheva kapia resulted in products with unexpected length, some of them with low intensity on the gel for visualization of the PCR products (Table 3). The amplified fragments from Oranzheva kapia, despite that their size differed from the expected, were cloned. Sequencing confirmed that these fragments were not related to *CrtZ*_chr03_ gene. Thus, we were not able to amplify fragments from this gene in the mutant, but we have successfully amplified them in the initial variety.

Table 3 is made according to Figure 4; thus, it covers only those reactions with a result in the mutant (they are of different lengths, but still have amplification also in the mutant). 

In contrast, amplification of fragments of the second β-carotene hydroxylase gene *Crt*Z_chr06_ produces fragments. These results suggest that the X-ray mutagenesis of initial variety Pazardzhiska kapia 794 most likely resulted in deletion of the *CrtZ*_ch03_ gene in the mutant variety Oranzheva kapia along with neighboring regions on chromosome 3 [35].

#### 3.2.2. Cloning and Sequencing of Both β-Carotene Hydroxylase Genes

The selected loci within the *Crt*Z_ch03_ gene cover its entire sequence. The lack of amplification in the mutant line at all positions suggests deletion of the gene. Corresponding PCR products from both lines were further cloned and sequenced (Figure 5). *Crt*Z_ch06_ gene fragments from all three lines were also amplified, cloned and sequenced.

In both control lines amplification resulted in products with expected lengths. In the mutant line, amplification products differed significantly from the predicted size (Figure 4). Sequences were identified comparing BLAST and the pepper genome database (http://peppersequence.genomics.cn/page/species/index.jsp). Several clones overlapping the entire sequence of the *CrtZ*_chr03_ gene from three pepper genotypes were sequenced, and consensus sequences for each genotype were generated. In initial and standard (control), Pazardzhishka kapia 794 and Kurtovska kapia 1 varieties, the obtained consensus sequences matched the known *Crt*Z_ch03_ gene from the NCBI database (accession numbers GU122940, GU122941, GU122942, GU122943, GU122944, GU122945, GU122946 and GU122940.1). Clones from the mutant variety Oranzheva kapia comprised non-*CrtZ_chr03_* related sequences, which suggest non-specific amplification.

The sequences of *CrtZ*_ch06_ in all the three genotypes—the mutant, the initial, and the control local variety—were explored. There were no differences between sequences in all three studied genotypes. The complete *CrtZ*_ch06_ sequence is presented in Appendix A. In the NCBI database, the only available data for the mRNA of β-carotene hydroxylase in chromosome 6 were accessions Y09722.1, XM_016720248.1 and XM_016720247.1. Hence, to determine and analyze the intron regions of the joined sequences, we conducted a BLAST analysis using the information available from the pepper genome database in http://peppersequence.genomics.cn/page/species/index.jsp (accessed on 10 June 2014). As a result of the comparison done with the pepper genome, we identified two differences in our sequences at positions 1413 G→A and 1517 T→A in the 4th intron established. Two more differences in our sequences were seen: at 1642, which was established with the gene’s cDNA sequence of the gene (accession number Y09722.1 in the NCBI), and another at position 455 in the pepper genome database.

The lack of *CrtZ*_ch03_ sequences in amplified products in the variety Oranzheva kapia further strengthens the hypothesis that X-ray irradiation deleted this gene.

#### 3.2.3. RT-PCR Analysis of Both (*CrtZ*_ch03_ and *CrtZ*_ch06_) β-Carotene Hydroxylase Genes

We analyzed the expression of both β-carotene hydroxylase genes in the initial variety Pazardzhishka kapia 794 and the mutant one Oranzheva kapia. Expression of *Crt*Z_ch03_ was detected only in the progenitor genotype Pazardzhishka kapia 794. No amplification was observed from the mutant genotype Oranzheva kapia (Figure 6).

These results demonstrate that the mutant variety Oranzheva kapia does not express mRNA for the *Crt*Z_ch03_ gene. In Figure 6, PCR reaction in lanes 2 and 3 was 219 bp, and the length of lane 4 was 341 bp, which matched the expected fragment lengths (measured in NCBI by using the sequence information of both hydroxylase genes).

Our data suggest that the accumulation of β-carotene in Oranzheva kapia is a result of the inactivation of the *Crt*Z_ch03_ gene. This in turn deactivates downstream metabolism of β-carotene. The pattern of amplification along with amplicon sequencing strongly suggest deletion of the entire *Crt*Z_ch03_ gene as a result of X-ray irradiation.

On the other hand, the *Crt*Z_ch06_ gene appeared unaffected and was expressed normally. Nevertheless, *Crt*Z_ch06_ cannot compensate for the inactivation of *Crt*Z_ch03_.

## 4. Discussion

The overall trend in the analyzed mutant fruits was the higher β-carotene concentration compared to the initial variety when the mutant plants were grown for green pepper up to the stage of technical maturity; the β-carotene concentration was considerably higher than in the initial, which demonstrates their high-added value. Data for the carotenoid spectra are consistent with a dominant inheritance of the red fruit color and that the gene responsible for the orange type of fruit pigmentation is a recessive one, stated in [36]. The fruits are intended for consumption (either green) as well as after reaching botanical maturity. The mutant appears to be a promising and significant source of β-carotene, similar to carrots, which are thought to be the main vegetable source for β-carotene in the human diet [37]. For comparison, β-carotene in the mutant pepper was 7.31 mg/100 g dry matter) and α-carotene was 0.67 mg/100 g dry matter (Table 2). These levels are similar to the concentrations found in the late sown carrot varieties [35,38].

The changes of the pigment composition were investigated during two phases of ripening for the varieties Oranzheva kapia and Pazardzhishka kapia 794. There were no existing comparative data for the changing concentrations of colored pigments for Oranzheva kapia (Figure 3). Based on the data in Table 1 and Table 2 it can be concluded that some keto carotenoid components significantly increase during the initial phases of ripening of the initial variety with red fruits. The ripening of the mutant is connected with the disappearance of the lutein and the relative decrease in other hydroxyxanthophylls. During advancing of ripening the concentrations of keto carotenoids increased significantly (Table 2). This can be explained by assuming that in Oranzheva kapia only the gene responsible for hydroxylation of the β-carotene to β-cryptoxanthin has mutated. The results validated the previous study that assessed the mutant gene determined higher β-carotene levels in the mutant genotypes [22,39]. We evaluated carotenoid levels in individual and group samples of a genotype in which the mutation was transferred [40]. The results validated the previous study that assessed all the genotypes containing the mutant gene that determines higher β-carotene levels (Table 1 and Table 2).

The variety Oranzheva kapia is characterized by high β-carotene concentrations of 7.31 mg/100 g dry matter, twice that of the initial variety Pazardzhishka kapia 794 with 3.76 mg/100 g dry matter. This is similar to the results of mutant lines originating from crosses of Oranzheva kapia with local pepper varieties that have red fruits [40].

An increase in β-carotene concentrations was observed during fruit maturation of the mutant lines. β-carotene levels were 2.25 times higher in green fruits and 1.94 times in botanically ripe fruits. β-carotene hydroxylase participates in conversion of β-carotene to β-cryptoxanthin [32]. These changes in pigment levels allow us to suggest that irradiation led to a mutation in the β-carotene hydroxylase gene(s). The resulting alteration in enzyme activity turns this product into the next stage of precursors in the metabolite chain. As a result, β-carotene hydroxylation to β-cryptoxanthin is suppressed, which leads to decreased levels of this pigment and zeaxanthin. Hence, we conclude a mutation occurred in the β-carotene hydroxylase gene caused by X-ray irradiation.The selected characteristic of the ß-carotene concentration data (Table 1 and Table 2) showed that this study’s genotypes for fresh consumption are promising as initial genetic material for MAS-related studies and for hybrid production [36,39].

In the carotenoid biosynthetic pathway, the lycopene is converted to γ-carotene, and in the next step the γ-carotene is converted to β-carotene (Figure 7). The mechanisms of the observed 7.5-fold increase in α-carotene in the mutant compared to the initial are not well understood. A plausible explanation is the redirection in γ-carotene to the synthesis of α-carotene [41]. Surprisingly, increased α-carotene and β-carotene levels correlate with decreased lutein concentration in the mutant. Its precursor is ε-carotene and, to a lesser extent, α-carotene. At this stage we can only speculate that increased α-carotene levels inhibit lycopene conversion into ε-carotene by β-carotene hydroxylase, thus reducing lutein.

The presence of α-carotene was detected in the mutant fruits but not in the initial variety. Presumably, α-carotene in the initial variety is rapidly converted to lutein, so the pigment is not detected in green fruits when measured by HPLC. In the mutant, lutein conversion is partially impaired. This reduces it to very low concentrations, but it is still present. The differences in the carotenes level might be explained by differences in the hydroxylation capacity of the beta-carotene hydroxylase acting on the two β-rings of β-carotene, and on the one β-ring of α-carotene. The hydroxylation of other carotene pigments, such as γ-carotene and probably δ-carotene, was also altered, but they were not measured in this study. We have no explanation for the decrease in lutein, which, unlike in the initial variety, is derived from α-carotene in the mutant.

The mutation is likely to affect the activity of the β-carotene hydroxylase. Gene deletion in turn affects the early stages of fruit development, as shown by the results of a comparative study of green fruits at technical maturity in the mutant and the initial genotypes. This leads to higher concentrations of β-carotene in the fruits of the mutant lines. In fruits at botanical maturity (red and orange) the levels of α-carotene and β-carotene in the mutant variety are higher than the initial variety. Since enzyme activity is disturbed, this does not allow hydroxylation of α-carotene and β-carotene and its transition to the next level of the biosynthetic pathway. As a result, the subsequent metabolites are quantified in significantly lower concentrations in the mutant.

Preliminary data mining for similar phenotypes had pointed to *CrtZ* as the most probable target gene [43,44]. The analysis of *CrtZ* expression by RT-PCR revealed that mRNA for *Crt*Z_ch06_ was presented in both genotypes, while *Crt*Z_ch03_ was expressed only in the initial and was absent in the mutant. *Crt*Z_ch06_ seemed to remain intact after mutagenesis; hence, the phenotype of Oranzheva kapia is solely the result of changes to *Crt*Z_ch03_ [35,39]. This proposition was supported by data from chemically induced mutation with EMS in *Crt*Z_ch03_ in the pepper variety Maor that led to A709G transition [45]. This transition resulted in enzyme inactivation and subsequent accumulation of β-carotene in fruits, with similar phenotype to Oranzheva kapia.

The expression data might reveal the roles of both β-carotene hydroxylase genes in carotenoid metabolism in pepper. Our data suggest that the key enzyme is β-carotene hydroxylase, encoded by *Crt*Z_ch03_. Its inactivation abolishes downward pathway steps. The second gene, *Crt*Z_ch06_, cannot replace *Crt*Z_ch03_ activity despite its normal expression levels [35]. The exact function of the *Crt*Z_ch06_ gene in plants needs further study.

Experiments with RT-PCR revealed that the *CrtZ*_ch03_ gene is not transcribed in the mutant but did not provide the reasons for this. In this particular mutation the inability to amplify genome fragments in the mutant, which was possible in the initial, led to the conclusion of locus deletion due to irradiation. Lack of amplification at several regions within the *CrtZ*_chr03_ locus suggests deletion of the entire gene. We are aware that point mutations, insertions or deletions outside the amplicons as well as translocation of the entire locus cannot be detected by this approach since the pattern will resemble that of the initial variety. The most plausible explanation of the inability to amplify a set of amplicons within the gene along with lack of transcription is the nonexistence of the target sequence within the genome.

## 5. Conclusions

The application of HPLC revealed that the orange fruit color in the mutant variety Oranzheva kapia correlates with high α- and β-carotene levels that are several times higher than in the initial fruits. While the level of β-carotene increased dramatically, the concentration of three xanthophylls (lutein, β-cryptoxanthin,- and zeaxanthin) decreased in the mutant fruit. Thus, consuming two to three fruits of the mutant pepper should meet the daily β-carotene requirement.

The performed PCR analysis allowed us to propose that the X-ray mutagenesis of the initial variety Pazardzhiska kapia 794 resulted in deletion of the *Crt*Z_ch03_ gene in the mutant variety Oranzheva kapia along with neighboring regions on chromosome 3. This deletion abolished the carotene biosynthetic pathway that led to the accumulation of α- and β-carotene and the resultant orange color of the mutant fruits. Henceforth, induced mutagenesis still remains an important tool for enhancing nutritional quality in the peppers. Last but not least, our results illustrate the unexpected and uncontrolled effects of induced mutagenesis on genome organization.

## Figures and Tables

**Figure 1 foods-10-01225-f001:**
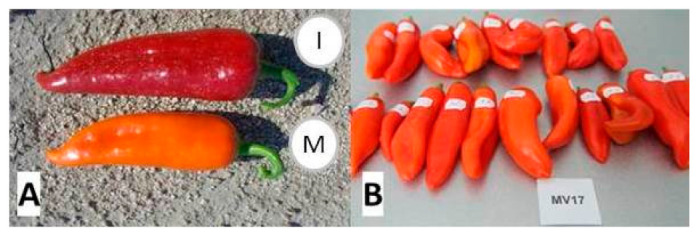
Botanical maturity fruit morphology of the two pepper varieties—(**A**) a red fruit characteristic for the variety Pazardzhishka kapia 794 (I) and an orange fruit of Oranzheva kapia (M), (**B**)—orange-fruited variety Oranzheva kapia (M) (20 fruits were collected and numbered for the HPLC analysis).

**Figure 2 foods-10-01225-f002:**
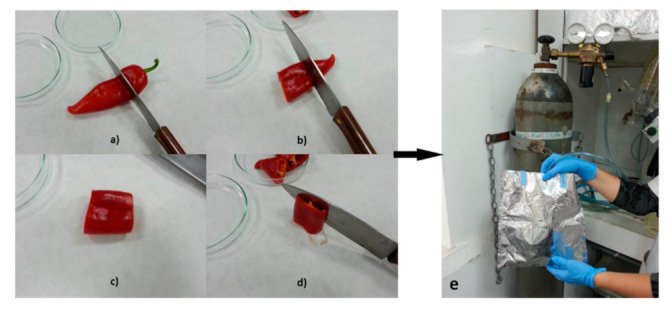
Overview of sample preparation. (**a**–**c**) Fruit was cut in three pieces; (**d**) middle part was used for analysis; (**e**) middle part was covered by nitrogen, lyophilized and enclosed in an aluminum container.

**Figure 3 foods-10-01225-f003:**
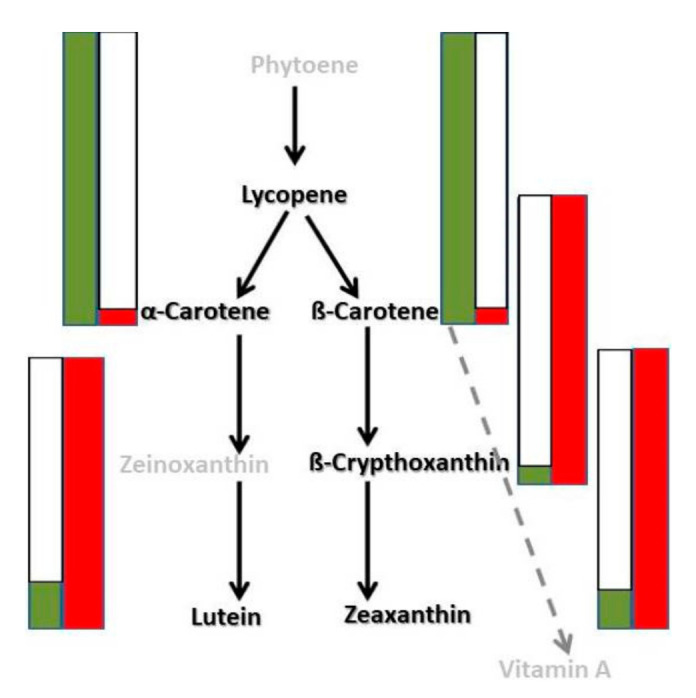
Schematic diagram of analyzed carotenoid substances in mutant and initial varieties in ripe fruits. Partial schematic representation of the carotenoid biosynthetic grid. Green bars—corresponding carotenoid levels in mutant variety, red bars—corresponding carotenoid levels in initial variety. Names of analyzed carotenoids in black color.

**Figure 4 foods-10-01225-f004:**
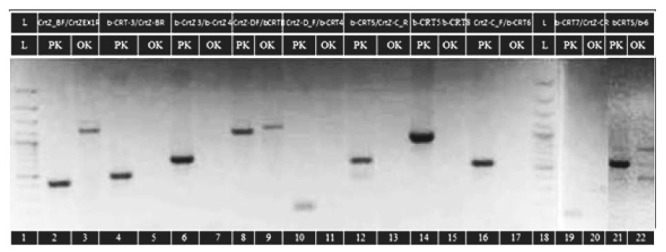
Amplification of ten loci within the *CrtZ*_ch03_ gene. Lanes 2, 4, 6, 8, 10, 12, 14, 16, 19 and 21 are from Pazardzhishka kapia 794 (PK). Lanes 3, 5, 7, 9, 11, 13, 15, 17, 20 and 22 are from the variety Oranzheva kapia (OK) mutant line. Lanes 1 and 18–100 bp are from the DNA Ladder.

**Figure 5 foods-10-01225-f005:**
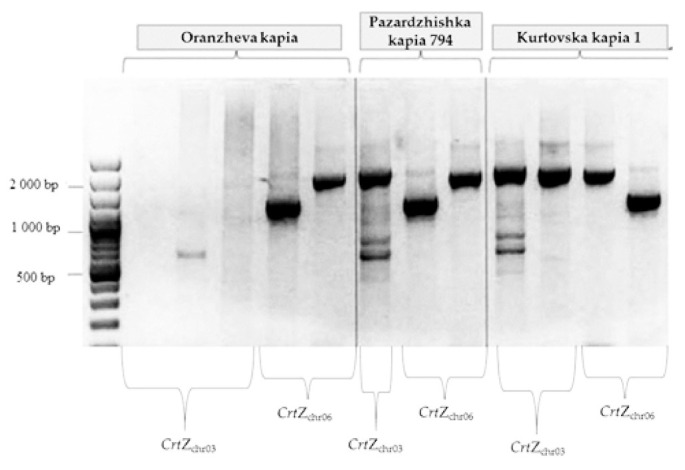
Amplification patterns of the loci, selected for sequencing. Lanes 2, 3, 4, 7 and 10 with primers positioned in *CrtZ*_ch03_. FW1_H2 1/b-CRT 8 R and expected length—2072 bp positioned in *CrtZ*_ch03_ gene. Lanes 5, 8 and 13 with primers FW1_H1 1/RE1_H1 1 and expected length—1212 bp positioned in *CrtZ*_ch06_. Lanes 6, 9 and 12 with primers FW1_H1 1/RE2_H1 1 and expected length—1863 bp positioned in *Crt*Z_ch06_. Lanes 2, 3, 4, 5 and 6—Oranzheva kapia. Lanes 7, 8 and 9—Pazardzhishka kapia 794, Lanes 10, 11, 12 and 13 in variety Kurtovska kapia 1. Lane 1–100 bp DNA Ladder.

**Figure 6 foods-10-01225-f006:**
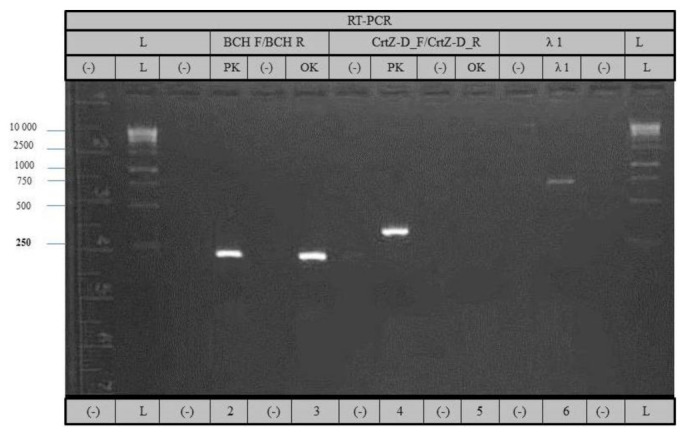
RT-PCR analysis of *Crt*Z_ch06_ with primers BCH F/BCH R (lane 2 and 3) and *Crt*Z_ch03_ with primers CrtZ-D–F/CrtZ–D–R (lanes 4 and 5). Pazardzhishka kapia 794 (lanes 2 and 4) and Oranzheva kapia (lanes 3 and 5). Lane 6—control amplification of elongation factor EF-1.

**Figure 7 foods-10-01225-f007:**
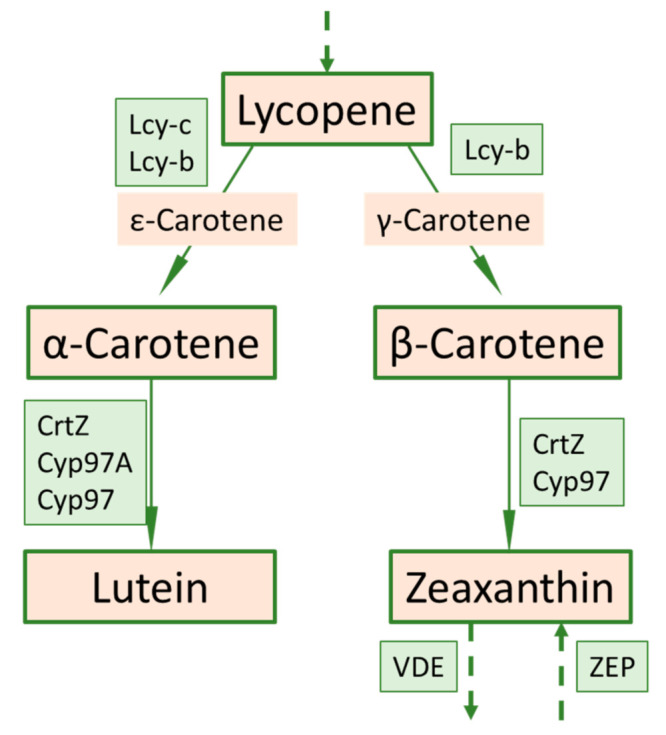
Part of the carotenoid biosynthetic pathway related to carotene metabolism including corresponding genes (in green boxes) redrawn according to [42].

**Table 1 foods-10-01225-t001:** Carotenoid concentrations (mg/100 g dry matter) of green fruits of Pazardzhishka kapia 794 (I) and Oranzheva kapia (M) varieties.

Genotypes	Genetic Relationships Between Genotypes	Lutein	β-Carotene
Pazardzhishka kapia 794	Initial	1.22 ± 0.21 *	0.32 ± 0.14 *
Oranzheva kapia	Mutant	0.74 ± 0.16 *	0.71 ± 0.23 *
Mutation Event	Alteration in M	1.65× decreased	2.25× increased

* Interval of confidence of average mean at *p* < 0.05.

**Table 2 foods-10-01225-t002:** Carotenoid concentrations in lyophilized samples from botanical ripe fruits (mg/100 g dry mater).

Genotypes	α-Carotene	Lutein	β-Carotene	BCX	Zea
Pazardzhishka kapia 794	0.09 ± 0.04 *	11.52 ± 3.76 *	3.76 ± 0.90 *	4.07 ± 1.66 *	18.46 ± 3.92 *
Oranzheva kapia	0.67 ± 0.21 *	1.37 ± 0.45 *	7.31 ± 0.98 *	0.000	2.67 ± 0.52 *
Mutation Event	7.49× higher	8.39× lower	1.94× higher	Not detected in M	6.90× lower

* Interval of confidence of average mean at *p* < 0.05.

**Table 3 foods-10-01225-t003:** Primer pairs used for amplification of selected loci within the *CrtZ*_chr03_ gene.

Primers for *CrtZ*_chr03_ Gene	Amplified Regions on Exons and Introns of the Gene	Position of Primers in the Genomic Sequence	Expected Fragments Length [bp]	Results from Amplifications of Genotypes	
				Initial	Mutant
CrtZ-B_F/ CrtZ_R EX1	E′1 + E′1	7/438	432	432	1300
b-CRT 3 F/ CrtZ-B_R	I′1 + E2 + I2 + E′3	483/970	488	488	500
b-CRT 3 F/ b-CRT 4 R	I′1 + E2 + I2 + E3 + I3 + E′4	483/1155	673	673	750
CrtZ-D_F/ b-CRT 8 R	E′3 + I3 + E4 + I4 + E5 + I5 + E6 + I6 + E′7	966/2088	1123	1123	1300
CrtZ-D_F/ b-CRT 4	E′3 + I3 + E′4	966/1155	190	190	200
b-CRT 5 F/ CrtZ-C_R	I′3 + E′4 + I4 + E5 + I5 + E′6	1127/1784	658	658	900
b-CRT 5 F/ b-CRT 8 R	I′3 + E′4 + I4 + E5 + I5 + E6 + I6 + E′7	1127/2088	962	962	1250
CrtZ-C_F/ b-CRT 6 R	E′4 + I4 + E5 + I′5	1163/1710	548	548	350
b-CRT 7 F/ CrtZ-C_R	I′5 + E′6	1602/1784	183	183	400
b-CRT 5 F/ b-CRT6 R	I′3 + E4 + I4 + E5 + I′5	1127/1710	584	584	750, 580, 450, 300

E—exon, I—intron, 1–4—number of exons/introns of the target gene.

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
