# Peer review of "Mutation Associated with Orange Fruit Color Increases Concentrations of β-Carotene in a Sweet Pepper Variety (Capsicum annuum L.)"

_foods, 2021, doi:10.3390/foods10061225_

Round 1

Reviewer 1 Report

It is difficult to approve this manuscript when there is no indication of any replication of quantitative experiments. Many, many other deficiencies exist as well. If there is replication and statistics can be calculated, then the objectives need to be focused, and the remainder of the manuscript should address those objectives. 

Author Response

Replies to the first review:

page 1, rоw 2.

Question: Does the mutation convert red to orange? Or is it associated with orange fruit color.

Reply: Apparently, this mutation is due to inactivation of beta-carotene hydroxylase gene. The breakage of carotenoid biosynthetic pathway at this step led to carotene accumulation. The lack of red-colored pigments and accumulation of carotene results in conversion of the initial red fruit color into orange one. Yes, the mutation converted the red fruit color of the initial variety into orange fruit color of the mutant. The associated increased carotene levels in the mutant are due to the mutation and are related to the orange fruit color. The orange fruit color is a result of the lack of conversion of the carotenes (alpha- and beta-carotenes are evaluated) into the next steps of the biosynthetic pathway. Thanks to your very reasonable suggestion, I am changing the title of the manuscript.

page 1, rоw 20.

Question: The success breeding does not rely on biochemical and molecular methods. Success can be enhanced, but it is not reliant on these methods.

Reply: Yes, we do agree. The sentence can be corrected:

Wrong sentence: The success of breeding programs relies on the efficient and integral application of different biochemical and molecular methods to characterize specific mutant alleles.

Corrected sentence: The success of breeding programs can be enhanced by the efficient and integral application of different biochemical and molecular methods to characterize specific mutant alleles.

page 1, rоw 21.

Question: Are you evaluating five levels or levels of 5 carotenoids. The latter I think. You should list the carotenoids.

Reply: levels of 5 carotenoids:  α-carotene, β-carotene, lutein, β-cryptoxanthin, zeaxanthin

Wrong sentence:  In this paper we present the evaluation of five carotenoid levels and the molecular characterization of a pepper mutant variety with high ß-carotene concentrations.

Corrected sentence:  In this paper we present the comparison of five carotenoid levels (α-carotene, β-carotene, lutein, β-cryptoxanthin, zeaxanthin) between a mutant and a corresponding initial pepper variety and the molecular characterization of the mutant variety with high ß-carotene concentrations.

page 1, rоw 24. (We found this error our-self.)

Wrong:  ………cost-effictive methods……….

Corrected:  ………cost-effective methods……….

page 1, rоw 25.

Wrong:  ………..…HPLC-DAD method……….r2 > 0.999………. six compounds ……………

Validation of the 25 HPLC-DAD method showed good linearity (r2 > 0.999) for the six compounds analyzed over a wide concentration range (0.03~5 μg / mL).

Corrected:  ………HPLC method………………R2 > 0.999………. five compounds………………..

Validation of the 25 HPLC-DAD method showed good linearity (R2 > 0.999) for the five compounds analyzed over a wide concentration range (0.03~5 μg / mL).

  1. 1, rоw 27.

Reviewer comment: Need more information here. Why were these genes identified? What are the likely substrates of the gene products?

Wrong:  PCR amplification, cloning, sequencing and expression of two hydroxylase genes were performed.

Corrected:  It was observed that fruit at both commercial and botanical maturity from mutant variety had greater α-carotene (0.6740 mg / 100 g dry matter) and β-carotene (0.711 and 7.3090 mg / 100 g dry matter, respectively) concentrations, to the initial variety. Two hydroxylase enzymes, converting α- and β-carotene, are known in pepper encoded by two hydroxylase genes. PCR amplification, cloning, sequencing and expression of two hydroxylase genes were performed.

  1. 1, row 33-34.

Wrong:  Peppers (Capsicum spp.) are a good source of dietary carotenoids, which has led to extensive

34 research on its carotenoid biochemistry [1].

Reviewer comment: their, to agree with peppers

Corrected:  Peppers (Capsicum spp.) are a good source of dietary carotenoids, which has led to extensive research on their carotenoid biochemistry [1].

  1. 2, r. 34

Reviewer comment: bred peppers

Reply: Pepper forms obtained in result of cultivation and breeding – this is a citation.

  1. 2, r. 42

Reviewer comment: Reference for: The economic importance of peppers is increasing, and the production is expanding.

Corrected: I am adding the reference of my review invited paper – Tomlekova N., 2010 …

The economic importance of peppers is increasing, and the production is expanding [5].

The reference added under number 5 demonstrates this increasing interest. The reference Tomlekova N., 2010 will replace the cited paper in the first submitted version of the paper – the old number [5].

Here is the new reference [5] that will replace the old [5]:

This one:

Tomlekova, N.B., Induced mutagenesis for crop improvement in Bulgaria. Plant Mutation Report, 2010, 2(2): 1-32. /invited paper/ ISSN: 1011-260X http://www.iaea.org/inis/collection/NCLCollectionStore/_Public/42/080/42080077.pdf?r=1

replaces the following one:

Tomlekova, N., Panchev, I., Yancheva, S., Todorova, V., Baudoin, J.P., Daskalov, S., Established molecular marker in pepper mutants with orange fruit colour and application in crop breeding for high beta-carotene. // Proceedings of the International Scientific Conference “Plant germplasm basis of the modern agriculture”, Plovdiv, Bulgaria, 2007, p. 379-383.

  1. 2, r. 42-44

Reviewer comment: Reference - Breeding programs have begun using complementary genomic approaches and efficient phenotypic and biochemical screening methods to identify and develop mutants with traits of interest such as vitamins and minerals.

Corrected: the reference added - Breeding programs have begun using complementary genomic approaches and efficient phenotypic and biochemical screening methods to identify and develop mutants with traits of interest such as vitamins and minerals [5].

  1. 2, r. 52-55

Reviewer comment: Reference - Within the past few years, a number of agronomically useful mutants have been induced and characterized, in addition to mutants that are valuable for genetic, cytological, biochemical, molecular and physiological studies. Many pepper varieties with improved nutritional value are available to meet human heath needs.

Corrected: the reference added - Within the past few years, a number of agronomically useful mutants have been induced and characterized, in addition to mutants that are valuable for genetic, cytological, biochemical, molecular and physiological studies [5].

  1. 2, r. 62-65

Reviewer comment:

So, I am halfway through the introduction, and now you tell me that we are dealing processed pepper, not fresh.

Consumer prefer red paprika (red is not required for grinding)

One of significant challenge in breeding peppers is the concentration of β-carotene in the fruits. When β-carotene concentration is high fruits are orange, but this reduces its commercial value since consumers prefer red which is required for grinding. Furthermore, the small quantities of red pepper used as a spice do not meet the β-carotene needs for human food. Genotypes bred for fresh consumption should have certain levels of β-carotene (provitamin A).

Corrected: I see, the sentences are not enough clear. I corrected the above text:

One of significant challenge in breeding peppers is the concentration of β-carotene in the fruits [13]. The richest natural source of β-carotene is the paprika for grinding. The small quantities of paprika used as a spice do not meet the β -carotene needs in human food. Since pepper for fresh consumption is consumed in big quantities, it is the main source, which provides the β -carotene in the human diet [14]. Thus, genotypes bred for fresh consumption should have certain levels of β -carotene (provitamin A).

Furthermore, when -carotene concentration is high, fruits are orange, but this reduces its commercial value since consumers prefer traditionally red pepper fruits for grinding.

  1. 2, r. 70

Reviewer comment: The red color is

Corrected: The fruit color is

  1. 2, r. 75

Reviewer comment: drung

Corrected: during

  1. 2, r. 75-76

Reviewer comment:  To be useful in breeding programs, a more detailed characterization of the mutant allele is required.

Corrected:  A more detailed characterization of the mutant allele is required.

  1. 3, r. 77-81

Reviewer comment:  You did not have to do this analysis to develop the hypothesis. It was obvious prior to your analysis that the mutation was likely lack of function, and thus there was high liklihood that downstream product concentrations were altered.

Answer: There are different changes of the biosynthetic pathway that occur leading to the accumulation of some carotenoids resulting into orange fruit color. The results from the analyses of the selected carotenoids can give idea about the changes in the genes encoding enzymes that convert the carotenoids into the next in the pathway. Some of the genes if mutated lead to such of alterations of the carotenoids levels are listed in the end of the section “Introduction”.

Corrected:  This paper’s goal is to biochemically evaluate carotenoid levels in parent and mutant sweet pepper genotypes.

  1. 3, r. 78-

Reviewer comment:  Not an hypothesis.

This allows us to develop a hypothesis for the genetic changes that results from induced mutagenesis, which boosts the health-related compounds in fruit. In the second stage of this study we present current progress in molecular analysis of the mutant variety with high β-carotene concentration.

Corrected:  

This reveals the genetic changes that result from induced mutagenesis, which boosts the carotene health-related compounds in fruit. In the second stage of this study we present current progress in molecular analysis of the mutant variety with high β-carotene concentration.

  1. 3, r. 85

Reviewer comment:  was to be corrected – were

parent / mutant was included in this study:

Corrected:  I would keep “was” because it is used for a couple of NIL/RIL

The corrected sentence will be:

The research was carried out on Capsicum annuum Linn. А couple of initial and the originating from it mutant was included in this study: Pazardzhishka kapia 794 and Oranzheva kapia. To assess target gene sequences, we also compare the local variety Kurtovska kapia 1.

  1. 3, r. 87-96

Reviewer comment:  This is not materials and methods. This belongs in the introduction. The authors did not do this work. What is Ms?

Corrected: The paragraphs were moved to the “Introduction” as it was recommended.

Answer to the reviewer:  The technology of obtaining the mutant from the initial genotype is described in details in the following paragraph. It is a standard technology and the 4 citations showed how it was obtained. In the section “Material and methods” we described shortly the plant material used for the study. To reply the question of the reviewer (1) I am describing the details of obtaining the couple RIL/NIL used in the study.

Ms are the mutant generations of the mutation breeding technology used to develop these mutant generations. After applying the irradiation (nuclear technology) by X-rays, mutations are generated in M1 and only dominant mutations are observed in M1. In M2 and M3 segregating generations the recessive mutations are selected according to the goal of the mutagenesis program (in this case – to generate mutations leading to the increase of carotene, because pepper is the major natural source for human diet of beta-carotene within all vegetables and the increase of this compound makes it valuable for the human diet. When a beneficial mutation is induced in the next stage it has to be fixed. The early mutant lines are back-crossed by the initial variety followed by selection. Selected are the mutants in which the orange fruit color (encoded by 2 recessive mutant alleles) is in presence. These RILs. Next generations up to M8 advanced mutant line are self-pollinated and a selection of plants with orange fruits was done. These are NIL lines. Thus, the advanced mutant line is RIL/NIL and it is homogenous with the initial except the mutant character. After developing the advanced M8 mutant line it is submitted for variety registration and released as mutant variety. This mutant line and the mutant variety developed (Oranzheva kapia) from the line are theoretically the same like the initial variety used for the irradiation (Pazardzhishka kapia 794) and differ each other only by the mutant trait: initial – with red fruits, mutant – orange.

  1. 3, r. 98-99

Reviewer comment:  Is it red, purple, green or what? Used for paprika? Where did you obtain seed.

Answer to the reviewer:  The seeds are from our collection of the Maritsa Vegetable Crops Research Institute. We have a separate unit for variety maintenance and seed production. We are maintaining our own varieties developed in the institute where the seed production is also performed. It is part of the collection.

Corrected sentence:  Kurtovska kapia 1 is a commonly grown local variety that has been developed by the method of continuous individual selection and maintained in the Maritsa Vegetable Crops Research Institute.

  1. 3, r. 100-101

Reviewer comment:  How did you grow the plants? How many seasons? Where did you grow the plants, etc. Leaf samples not leafs samples. What was the right time?

Corrected sentence:  All fruit and leaf samples from mutant and initial genotypes were lyophilized and stored in the dark in covered aluminum containers at 4 oC for preservation and to be analzed at the right time.

Growth conditions and harvested portions

Ten plants of each genotype were grown in the field at the Maritsa Vegetable Crops Research Institute, Plovdiv, Bulgaria. The plants were grown two successive years in furrows following conventional practice for mid-early pepper production in this region. For analyses of carotenoids, twenty green fruit from the initial and twenty - from the mutant genotype were analysed at commercial maturity, and twenty red fruit from the initial and twenty orange fruit from the mutant variety were analysed at botanical maturity. Fruit of each maturity x genotype were of uniform size and colour.

  1. 4, r. 125

Reviewer comment:  heater not thermostat

Corrected sentence:  heater

  1. 4, r. 131

Reviewer comment:  What proportion. This appears to be a reverse phase column. What were the beginning and ending proportions of A and B, Furthermore, usually A is starting conditions and B is ending, but the polarity of B > A  as cited.

The sentence was:

The mobile phase consisted of acetonitrile, methanol, ethyl acetate [11] of acetonitrile – methanol (95:5, v/v) (eluent A) and acetonitrile – methanol – ethyl acetate (60:20:20, v/v/v) (eluent B), containing 0.1 % BHT as an antioxidant and 0.05 % trietylamine (TEA). A concave gradient was applied for 20 min to maintain this proportion until the end of the run.

Corrected sentence: 

The mobile phase consisted of acetonitrile, methanol, ethyl acetate [11] of acetonitrile – methanol (95:5, v/v) (eluent A) and acetonitrile – methanol – ethyl acetate (60:20:20, v/v/v) (eluent B), containing 0.1 % BHT as an antioxidant and 0.05 % trietylamine (TEA). A concave gradient from 100 % A to 100 % B was applied for 15 min until the end of the run.

  1. 4, r. 133-134

Reviewer comment:  450 nm is not UV detection.

Wrong sentence:  UV-Vis detection was done at a wavelength of 450 nm.

Corrected sentence:  Elution profile was recorded by UV-Vis detection module set at 450 nm.

  1. 4, r. 143-153

Reviewer comment:  Results, not Material and Methods

Corrected sentence:  The three subsections were moved to the Results section of the manuscript as recommended.

  1. 5, r. 164

Reviewer comment:  Freeze dried, right?

Genomic DNA was prepared from young leaves according…

Corrected sentence: 

Genomic DNA was isolated from freeze-dried young leaves according to [25, 26].

  1. 5, r. 166-168

Reviewer comment:  Unclear how simply running lamda DNA on a gel would lead to quantitation.

Lambda DNA (Thermo Fisher Scientific, Lithuania) was used to determine for DNA quantity on 1 % LE agarose gels (Lonza, USA) with Ethidium bromide (VWR International, Austria).

Corrected sentence: 

DNA concentration in the samples was estimated by comparing band intensity of probes with lambda DNA of known concentrations (50 ng / μL, 20 ng / μL, 10 ng / μL and 5 ng / μL). 3 μL of each dilution were loaded per lane.

  1. 5, r. 169

Reviewer comment:  Was this needed for freeze dried samples?

Total RNA was isolated from the pericarp after grinding in liquid nitrogen.

Answer to the reviewer: Fresh samples were freeze in liquid nitrogen and stored at -80 oC. Total RNA was isolated from the pericarp after grinding in liquid nitrogen.

  1. 5, r. 174

Reviewer comment:  Incomplete description of how this was accomplished. You describe what you did, but not how.

Gene-specific primers for β-carotene hydroxylase genes were located on chromosomes 3 and 6 of the pepper genome (Supplementary Information 1, Table S1).

Answer to the reviewer:

CrtZchr03 and CrtZchr06 cDNA regions were amplified from total RNA using One-step RT-PCR kit (Qiagen, USA) according to the manufacturer instructions. Briefly, about 1 μg total RNA was mixed with 0.4 μM primers (CrtZ-D_f/CrtZ-D_R for CrtZchr03 gene or BCH1-F/BCH1-R for CrtZchr06 gene). The mix was incubated for 15 min at 95 oC and cooled on ice. 5 μL One-step RT-PCR buffer, 1 μL 10 mM dNTP and 1 μL One-step RT-PCR enzyme mix were added along with ddH20 to a final volume of 25 μL.

Reverse transcription was performed for 25 min at 20 oC and then denatured for 5 min at 94 oC. Then 35 amplification cycles were performed as follows: 1 min at 94 oC, 50 sec at 55 oC, 2 min at 72 oC followed by final extension for 7 min at 72 oC.

The Elongation factor 1 alpha (EF1 - At1g07940) was used as positive control with primers EF1-F: ATT GTG GTC ATT GGY CAY GT and EF1-R: CCA ATC TTG TAV ACA TCC TG (Cho et al., 1995, Kozera and Rapacz, 2013) under same amplification conditions. Amplification products were separated on 1.5 % agarose gel in TAE containing 0.5 μg / mL ethidium bromide. Product length was estimated by comparing with GeneRuler 1 kb DNA Ladder (Thermo Scientific).

  1. 5, r. 178

Reviewer comment:  No description of how cloning and sequencing were accomplished.

Gene cloning and sequencing

Answer to the reviewer:

Amplified fragments were cloned into pTZ57R plasmid vector using InsTAclone kit (Thermo Scientific) and transformed in E. coli DH5alpha. Plasmids from positive clones were isolated and sent for sequencing (StarSeq GmbH, Germany).

  1. 5, r. 181-182

Reviewer comment:  How did you managing to analyze a clone in a mutant genotype?

Clones derived from the variety Oranzheva kapia were analyzed by primer sets CrtZ–D_F/b-CRT 8, CrtZ-E/b-CRT 4 and b-CRT 7/CrtZ-C_R in the mutant genotype of Oranzheva kapia.

Answer to the reviewer:

We have cloned the amplified fragments despite that their size differs from the expected. Sequencing confirmed that these fragments are not related to CrtZchr03 gene. We were not able to amplify fragments from this gene in the mutant while we have successfully amplified them in the initial variety.

  1. 6, r. 225

All local varieties had mature red fruits

  1. 6, r. 227-229

Reviewer comment: Provide the data, with standard errors.

Corrected:

In Material and Methods:

The average mean and the significance interval of the studied characteristics were calculated by SPSS (descriptive statistics). Carotenoid levels were evaluated in individual and group samples [32].

In Results:

When Pazardzhiska kapia 794 and Oranzheva kapia were compared the intervals of average values of the quantified carotenoid substances did not have common point and did not intersect. The obtained data from the confidence intervals of the average values proved that the differences between initial and mutant varieties were statistically significant.

In Discussion:

The results validated the previous study that assessed the mutant gene determined higher β-carotene levels in the mutant genotypes [21]. We evaluated carotenoid levels in individual and group samples [32]. The results validated the previous study that assessed all the genotypes containing the mutant gene that determines higher β-carotene levels.

  1. 6, r. 227-229

Reviewer comment:  According to materials and methods, the run was 20 minutes long. The author only reports ~15 minutes of response. Please explain. The area of the graph beyond 16 minutes is empty space, What concentrations were evaluated? What is mAU?

Answer to the reviewer:

The run time was 15 minutes followed by 5 min. column washing.

  1. 7, r. 256

Reviewer comment:  Table 1. Determination of carotenoid concentration of dry matter (mg / 100 g) of green fruits of Pazardzhishka kapia 794 (parent) and Oranzheva kapia (mutant) varieties (mg / 100 g).

Answer to the reviewer:  Table 1. Carotenoid concentration of dry matter (mg / 100 g) of green fruits of Pazardzhishka kapia 794 (parent) and Oranzheva kapia (mutant) varieties (mg / 100 g).

Reviewer comment:  No replication?  Where are standard errors, etc?

Answer to the reviewer:  Table 1.

added

  1. 8, r. 262

Reviewer comment:  no measure of dispersion

Table 2. Determination of carotenoid concentration in dry matter lyophilized samples from botanical

ripe fruits (mg / 100 g).

  1. 8, r. 267

Reviewer comment:  Figure is fuzzy. What does the height of the red and green bars indicate?

Schematic diagram of analysed carotenoid substances in mutant and parent varieties in ripe

fruits. Partial schematic representation of the carotenoid biosynthetic grid.

Answer to the reviewer: 

Figure 3. Schematic diagram of analysed carotenoid substances in mutant and initial varieties in ripe fruits. Partial schematic representation of the carotenoid biosynthetic grid. Green bars – analysed samples from mutant variety, red bars – analysed samples from initial variety.

  1. 8, r. 269

Reviewer comment:  Figure 3 says nothing about absence of lycopene in the samples. Figure 3 only shows standards, not fruit extracts. I can only conclude that your method is incapable of detecting lycopene.

Answer to the reviewer: 

The figure 3 shows analysed samples. Lycopene was not detected in pepper samples. No information in the bibliography about the detection of lycopene in pepper fruits.

  1. 8, r. 269

Reviewer comment:  Incomplete description of figure.

Figure 5. Amplification patterns of the loci, selected for sequencing.

Answer to the reviewer:  ??? Figure description corrected

  1. 11, r. 347-353

Reviewer comment:  Belongs to Introductory material

The orange-coloured variety Oranzheva kapia was developed via 120 Gy X – ray dose application to Pazardzhishka kapia 794, commonly used local variety in Bulgaria. The mutant plants were back-crossed with parent (3 generations) and self-pollinated for a total of 8 generations, thus generating NIL/RIL [20].

Developed as NIL / RIL, the mutant variety Oranzheva kapia was genetically very close to the parent one Pazardzhishka kapia 794. Their differences are only due to the mutation in Oranzheva kapia.

Answer to the reviewer:  The paragraphs are moved to Introductory section.

  1. 11, r. 357-358

Reviewer comment:  What? I thought we were aimed at paprika production,

Answer to the reviewer: 

The sentence: The fruits are primarily intended for consumption while green, underlines that the pepper is sweet pepper, not paprika for grinding, not red pepper for grinding

Inserted in the beginning of the Introduction is the following: “sweet pepper ‘kapia’ type intended for fresh consumption”, and which would remove any ambiguity. The details are related to Oranzheva kapia and Pazardzhishka kapia 794 varieties – subjects of study.

It is also added about Kurtovska kapia 1 - It also belongs to ‘kapia’ type a sweet pepper.

The capsanthin content is of importance for the paprika for grinding. Here we work with sweet pepper for fresh consumption, that may be also processed.

  1. 11-12, r. 362-363

Reviewer comment:  Reference

Answer to the reviewer:  added [37]

Added ‘carrot’

  1. 12, r. 368

Reviewer comment:  Do you mean pigment composition?

The changes of the pigment spectrum were

Corrected: 

The changes of the pigment composition were

  1. 12, r. 371

Reviewer comment:  Not without statistics.

Significantly

Corrected:  Statistics added

  1. 12, r. 377

Reviewer comment:  Really, you have confidence in 4 decimal places?

Corrected:  They are decreased to three decimal places in the manuscript.

  1. 12, r. 379

Reviewer comment:  NO, it is similar to other lines carrying the orange phenotype and mutation present in O k.

This is similar to the results of mutant lines originating from crosses of Oranzheva kapia

Answer to the reviewer:  Not every orange phenotype will be with high beta-carotene. In some cases the red colour can be less intensive (the carotenoids with red colour will contribute less) and together with the yellow colour it will result into orange frui colour. In this case the beta-carotene (the orange pigment) is not increased and it is not the reason that determines the orange fruit. An eventual mutation in CCS may give this effect.

  1. 12, r. 393

Reviewer comment:  Should have been presented in introduction.

Corrected:  If this statement is in introduction the following will not be well understood:

The mechanisms of the observed 7.5 fold increase of α-carotene in the mutant compared to the parent are not well understood.

  1. 13, r. 410

Reviewer comment:  What are parent mutants?

Answer to the reviewer:  parent is replaced in the manuscript into initial mutants

parent mutants removed and ‘initial fruits’ used.

  1. 13, r. 421

Reviewer comment:  Why?

The mutation is likely to affect the presence of the enzyme or the regulation of .-carotene

hydroxylase enzyme activity.

Answer to the reviewer:    These two cases are the only logical. I don’t know other opportunity that results in a similar phenotype.

  1. 13, r. 421

Reviewer comment:  early stages

Answer to the reviewer:    ‘early stages of plant development’ added to the existing text

  1. 13, r. 431

Reviewer comment:  Probably should not have concluded earlier that it was a deletion. What kinds of mutations are expected with X-ray?

Sentence: is an event leading to the deletion, inversion, translocation or insertion of a retrotransposon into the sequence of the target gene of the mutant, occurring frequently after physical mutagen treatment.

Answer to the reviewer:  After applying Rӧ-rays irradiation the mutations are frequently deletions of about 50 bp fragment(an old statistics). However, every type of mutation can occur after X-rays treatment. The mutation cannot be predicted. can occur

  1. 13, r. 433

Reviewer comment:  Relevance of this statement

This study’s results were obtained after amplification, cloning and sequencing of the two .-carotene hydroxylase genes.

Answer to the reviewer:  Not relevant. Removed.

  1. 13, r. 440

Reviewer comment:  What is this, explain?

Answer to the reviewer:    A709G transition. - this is the exact nucleotide position along the length of the target gene affected by the mutation that is a transition - A replaced by G at 709 nd position. This sign explains what nucleotide is replaced by what as a mutation event.

  1. 14, r. 451

Reviewer comment:  Seems to me this paper should have been evident in the introduction.

Removed sentence: The lack of amplification at several points within the gene eliminates translocation and inversion as possible reasons.

Corrected sentence: Lack of amplification at several regions within CrtZchr03 locus suggests gene deletion instead of translocation or inversion.

Answer to the reviewer:    This means that our Results and Discussion are very logical and well understood. We did not know this result all these before the amplification, cloning and sequencing. There are so many different options that could be, instead of the ones found here.

Probably should have indicated earlier why this line of research was pursued.

The lack of amplification at several points within the gene eliminates translocation and inversion as possible reasons. Insertion of a mobile element or another fragment is also eliminated.

  1. 14, r. 457-458

Modern breeding programs need detailed characterizations of available germplasm. This

includes not only genotyping and markers but also characterization of mutations for better

understanding of processes in the plant and expected output.

I agree. And any phenotypic characterization should include estimates of dispersion.

  1. 5, r. 461

One way to meet breeder’s need is to carefully

design experiments and to select proper tools from the enormous molecular biology arsenal.

  1. 5, r. 463

Corrected Conclusions:

The application of HPLC revealed that the orange fruit color in the mutant variety Oranzheva kapia correlates with high α- and β-carotene levels that are several times higher than in the parent fruits. While the level of β-carotene increased dramatically, the concentration of three xanthophylls (lutein, β-cryptoxanthin, and zeaxanthin) decreased in the mutant fruit. Thus, consuming two to three fruits of the mutant pepper should meet the daily β-carotene requirement.

The performed PCR analysis allowed us to propose that the X-ray mutagenesis of parent variety Pazardzhiska kapia 794 resulted in deletion of the CrtZch03 gene in the mutant variety Oranzheva kapia along with neighboring regions on chromosome 3. This deletion abolished the carotene biosynthetic pathway that led to the accumulation of α- and β-carotene and the resultant orange color of the mutant fruits. Henceforth, induced mutagenesis still remains an important tool for enhancing nutritional quality in the peppers.

Replies to the first review:

page 1, rоw 2.

Question: Does the mutation convert red to orange? Or is it associated with orange fruit color.

Reply: Apparently, this mutation is due to inactivation of beta-carotene hydroxylase gene. The breakage of carotenoid biosynthetic pathway at this step led to carotene accumulation. The lack of red-colored pigments and accumulation of carotene results in conversion of the initial red fruit color into orange one. Yes, the mutation converted the red fruit color of the initial variety into orange fruit color of the mutant. The associated increased carotene levels in the mutant are due to the mutation and are related to the orange fruit color. The orange fruit color is a result of the lack of conversion of the carotenes (alpha- and beta-carotenes are evaluated) into the next steps of the biosynthetic pathway. Thanks to your very reasonable suggestion, I am changing the title of the manuscript.

page 1, rоw 20.

Question: The success breeding does not rely on biochemical and molecular methods. Success can be enhanced, but it is not reliant on these methods.

Reply: Yes, we do agree. The sentence can be corrected:

Wrong sentence: The success of breeding programs relies on the efficient and integral application of different biochemical and molecular methods to characterize specific mutant alleles.

Corrected sentence: The success of breeding programs can be enhanced by the efficient and integral application of different biochemical and molecular methods to characterize specific mutant alleles.

page 1, rоw 21.

Question: Are you evaluating five levels or levels of 5 carotenoids. The latter I think. You should list the carotenoids.

Reply: levels of 5 carotenoids:  α-carotene, β-carotene, lutein, β-cryptoxanthin, zeaxanthin

Wrong sentence:  In this paper we present the evaluation of five carotenoid levels and the molecular characterization of a pepper mutant variety with high ß-carotene concentrations.

Corrected sentence:  In this paper we present the comparison of five carotenoid levels (α-carotene, β-carotene, lutein, β-cryptoxanthin, zeaxanthin) between a mutant and a corresponding initial pepper variety and the molecular characterization of the mutant variety with high ß-carotene concentrations.

page 1, rоw 24. (We found this error our-self.)

Wrong:  ………cost-effictive methods……….

Corrected:  ………cost-effective methods……….

page 1, rоw 25.

Wrong:  ………..…HPLC-DAD method……….r2 > 0.999………. six compounds ……………

Validation of the 25 HPLC-DAD method showed good linearity (r2 > 0.999) for the six compounds analyzed over a wide concentration range (0.03~5 μg / mL).

Corrected:  ………HPLC method………………R2 > 0.999………. five compounds………………..

Validation of the 25 HPLC-DAD method showed good linearity (R2 > 0.999) for the five compounds analyzed over a wide concentration range (0.03~5 μg / mL).

  1. 1, rоw 27.

Reviewer comment: Need more information here. Why were these genes identified? What are the likely substrates of the gene products?

Wrong:  PCR amplification, cloning, sequencing and expression of two hydroxylase genes were performed.

Corrected:  It was observed that fruit at both commercial and botanical maturity from mutant variety had greater α-carotene (0.6740 mg / 100 g dry matter) and β-carotene (0.711 and 7.3090 mg / 100 g dry matter, respectively) concentrations, to the initial variety. Two hydroxylase enzymes, converting α- and β-carotene, are known in pepper encoded by two hydroxylase genes. PCR amplification, cloning, sequencing and expression of two hydroxylase genes were performed.

  1. 1, row 33-34.

Wrong:  Peppers (Capsicum spp.) are a good source of dietary carotenoids, which has led to extensive

34 research on its carotenoid biochemistry [1].

Reviewer comment: their, to agree with peppers

Corrected:  Peppers (Capsicum spp.) are a good source of dietary carotenoids, which has led to extensive research on their carotenoid biochemistry [1].

  1. 2, r. 34

Reviewer comment: bred peppers

Reply: Pepper forms obtained in result of cultivation and breeding – this is a citation.

  1. 2, r. 42

Reviewer comment: Reference for: The economic importance of peppers is increasing, and the production is expanding.

Corrected: I am adding the reference of my review invited paper – Tomlekova N., 2010 …

The economic importance of peppers is increasing, and the production is expanding [5].

The reference added under number 5 demonstrates this increasing interest. The reference Tomlekova N., 2010 will replace the cited paper in the first submitted version of the paper – the old number [5].

Here is the new reference [5] that will replace the old [5]:

This one:

Tomlekova, N.B., Induced mutagenesis for crop improvement in Bulgaria. Plant Mutation Report, 2010, 2(2): 1-32. /invited paper/ ISSN: 1011-260X http://www.iaea.org/inis/collection/NCLCollectionStore/_Public/42/080/42080077.pdf?r=1

replaces the following one:

Tomlekova, N., Panchev, I., Yancheva, S., Todorova, V., Baudoin, J.P., Daskalov, S., Established molecular marker in pepper mutants with orange fruit colour and application in crop breeding for high beta-carotene. // Proceedings of the International Scientific Conference “Plant germplasm basis of the modern agriculture”, Plovdiv, Bulgaria, 2007, p. 379-383.

  1. 2, r. 42-44

Reviewer comment: Reference - Breeding programs have begun using complementary genomic approaches and efficient phenotypic and biochemical screening methods to identify and develop mutants with traits of interest such as vitamins and minerals.

Corrected: the reference added - Breeding programs have begun using complementary genomic approaches and efficient phenotypic and biochemical screening methods to identify and develop mutants with traits of interest such as vitamins and minerals [5].

  1. 2, r. 52-55

Reviewer comment: Reference - Within the past few years, a number of agronomically useful mutants have been induced and characterized, in addition to mutants that are valuable for genetic, cytological, biochemical, molecular and physiological studies. Many pepper varieties with improved nutritional value are available to meet human heath needs.

Corrected: the reference added - Within the past few years, a number of agronomically useful mutants have been induced and characterized, in addition to mutants that are valuable for genetic, cytological, biochemical, molecular and physiological studies [5].

  1. 2, r. 62-65

Reviewer comment:

So, I am halfway through the introduction, and now you tell me that we are dealing processed pepper, not fresh.

Consumer prefer red paprika (red is not required for grinding)

One of significant challenge in breeding peppers is the concentration of β-carotene in the fruits. When β-carotene concentration is high fruits are orange, but this reduces its commercial value since consumers prefer red which is required for grinding. Furthermore, the small quantities of red pepper used as a spice do not meet the β-carotene needs for human food. Genotypes bred for fresh consumption should have certain levels of β-carotene (provitamin A).

Corrected: I see, the sentences are not enough clear. I corrected the above text:

One of significant challenge in breeding peppers is the concentration of β-carotene in the fruits [13]. The richest natural source of β-carotene is the paprika for grinding. The small quantities of paprika used as a spice do not meet the β -carotene needs in human food. Since pepper for fresh consumption is consumed in big quantities, it is the main source, which provides the β -carotene in the human diet [14]. Thus, genotypes bred for fresh consumption should have certain levels of β -carotene (provitamin A).

Furthermore, when -carotene concentration is high, fruits are orange, but this reduces its commercial value since consumers prefer traditionally red pepper fruits for grinding.

  1. 2, r. 70

Reviewer comment: The red color is

Corrected: The fruit color is

  1. 2, r. 75

Reviewer comment: drung

Corrected: during

  1. 2, r. 75-76

Reviewer comment:  To be useful in breeding programs, a more detailed characterization of the mutant allele is required.

Corrected:  A more detailed characterization of the mutant allele is required.

  1. 3, r. 77-81

Reviewer comment:  You did not have to do this analysis to develop the hypothesis. It was obvious prior to your analysis that the mutation was likely lack of function, and thus there was high liklihood that downstream product concentrations were altered.

Answer: There are different changes of the biosynthetic pathway that occur leading to the accumulation of some carotenoids resulting into orange fruit color. The results from the analyses of the selected carotenoids can give idea about the changes in the genes encoding enzymes that convert the carotenoids into the next in the pathway. Some of the genes if mutated lead to such of alterations of the carotenoids levels are listed in the end of the section “Introduction”.

Corrected:  This paper’s goal is to biochemically evaluate carotenoid levels in parent and mutant sweet pepper genotypes.

  1. 3, r. 78-

Reviewer comment:  Not an hypothesis.

This allows us to develop a hypothesis for the genetic changes that results from induced mutagenesis, which boosts the health-related compounds in fruit. In the second stage of this study we present current progress in molecular analysis of the mutant variety with high β-carotene concentration.

Corrected:  

This reveals the genetic changes that result from induced mutagenesis, which boosts the carotene health-related compounds in fruit. In the second stage of this study we present current progress in molecular analysis of the mutant variety with high β-carotene concentration.

  1. 3, r. 85

Reviewer comment:  was to be corrected – were

parent / mutant was included in this study:

Corrected:  I would keep “was” because it is used for a couple of NIL/RIL

The corrected sentence will be:

The research was carried out on Capsicum annuum Linn. А couple of initial and the originating from it mutant was included in this study: Pazardzhishka kapia 794 and Oranzheva kapia. To assess target gene sequences, we also compare the local variety Kurtovska kapia 1.

  1. 3, r. 87-96

Reviewer comment:  This is not materials and methods. This belongs in the introduction. The authors did not do this work. What is Ms?

Corrected: The paragraphs were moved to the “Introduction” as it was recommended.

Answer to the reviewer:  The technology of obtaining the mutant from the initial genotype is described in details in the following paragraph. It is a standard technology and the 4 citations showed how it was obtained. In the section “Material and methods” we described shortly the plant material used for the study. To reply the question of the reviewer (1) I am describing the details of obtaining the couple RIL/NIL used in the study.

Ms are the mutant generations of the mutation breeding technology used to develop these mutant generations. After applying the irradiation (nuclear technology) by X-rays, mutations are generated in M1 and only dominant mutations are observed in M1. In M2 and M3 segregating generations the recessive mutations are selected according to the goal of the mutagenesis program (in this case – to generate mutations leading to the increase of carotene, because pepper is the major natural source for human diet of beta-carotene within all vegetables and the increase of this compound makes it valuable for the human diet. When a beneficial mutation is induced in the next stage it has to be fixed. The early mutant lines are back-crossed by the initial variety followed by selection. Selected are the mutants in which the orange fruit color (encoded by 2 recessive mutant alleles) is in presence. These RILs. Next generations up to M8 advanced mutant line are self-pollinated and a selection of plants with orange fruits was done. These are NIL lines. Thus, the advanced mutant line is RIL/NIL and it is homogenous with the initial except the mutant character. After developing the advanced M8 mutant line it is submitted for variety registration and released as mutant variety. This mutant line and the mutant variety developed (Oranzheva kapia) from the line are theoretically the same like the initial variety used for the irradiation (Pazardzhishka kapia 794) and differ each other only by the mutant trait: initial – with red fruits, mutant – orange.

  1. 3, r. 98-99

Reviewer comment:  Is it red, purple, green or what? Used for paprika? Where did you obtain seed.

Answer to the reviewer:  The seeds are from our collection of the Maritsa Vegetable Crops Research Institute. We have a separate unit for variety maintenance and seed production. We are maintaining our own varieties developed in the institute where the seed production is also performed. It is part of the collection.

Corrected sentence:  Kurtovska kapia 1 is a commonly grown local variety that has been developed by the method of continuous individual selection and maintained in the Maritsa Vegetable Crops Research Institute.

  1. 3, r. 100-101

Reviewer comment:  How did you grow the plants? How many seasons? Where did you grow the plants, etc. Leaf samples not leafs samples. What was the right time?

Corrected sentence:  All fruit and leaf samples from mutant and initial genotypes were lyophilized and stored in the dark in covered aluminum containers at 4 oC for preservation and to be analzed at the right time.

Growth conditions and harvested portions

Ten plants of each genotype were grown in the field at the Maritsa Vegetable Crops Research Institute, Plovdiv, Bulgaria. The plants were grown two successive years in furrows following conventional practice for mid-early pepper production in this region. For analyses of carotenoids, twenty green fruit from the initial and twenty - from the mutant genotype were analysed at commercial maturity, and twenty red fruit from the initial and twenty orange fruit from the mutant variety were analysed at botanical maturity. Fruit of each maturity x genotype were of uniform size and colour.

  1. 4, r. 125

Reviewer comment:  heater not thermostat

Corrected sentence:  heater

  1. 4, r. 131

Reviewer comment:  What proportion. This appears to be a reverse phase column. What were the beginning and ending proportions of A and B, Furthermore, usually A is starting conditions and B is ending, but the polarity of B > A  as cited.

The sentence was:

The mobile phase consisted of acetonitrile, methanol, ethyl acetate [11] of acetonitrile – methanol (95:5, v/v) (eluent A) and acetonitrile – methanol – ethyl acetate (60:20:20, v/v/v) (eluent B), containing 0.1 % BHT as an antioxidant and 0.05 % trietylamine (TEA). A concave gradient was applied for 20 min to maintain this proportion until the end of the run.

Corrected sentence: 

The mobile phase consisted of acetonitrile, methanol, ethyl acetate [11] of acetonitrile – methanol (95:5, v/v) (eluent A) and acetonitrile – methanol – ethyl acetate (60:20:20, v/v/v) (eluent B), containing 0.1 % BHT as an antioxidant and 0.05 % trietylamine (TEA). A concave gradient from 100 % A to 100 % B was applied for 15 min until the end of the run.

  1. 4, r. 133-134

Reviewer comment:  450 nm is not UV detection.

Wrong sentence:  UV-Vis detection was done at a wavelength of 450 nm.

Corrected sentence:  Elution profile was recorded by UV-Vis detection module set at 450 nm.

  1. 4, r. 143-153

Reviewer comment:  Results, not Material and Methods

Corrected sentence:  The three subsections were moved to the Results section of the manuscript as recommended.

  1. 5, r. 164

Reviewer comment:  Freeze dried, right?

Genomic DNA was prepared from young leaves according…

Corrected sentence: 

Genomic DNA was isolated from freeze-dried young leaves according to [25, 26].

  1. 5, r. 166-168

Reviewer comment:  Unclear how simply running lamda DNA on a gel would lead to quantitation.

Lambda DNA (Thermo Fisher Scientific, Lithuania) was used to determine for DNA quantity on 1 % LE agarose gels (Lonza, USA) with Ethidium bromide (VWR International, Austria).

Corrected sentence: 

DNA concentration in the samples was estimated by comparing band intensity of probes with lambda DNA of known concentrations (50 ng / μL, 20 ng / μL, 10 ng / μL and 5 ng / μL). 3 μL of each dilution were loaded per lane.

  1. 5, r. 169

Reviewer comment:  Was this needed for freeze dried samples?

Total RNA was isolated from the pericarp after grinding in liquid nitrogen.

Answer to the reviewer: Fresh samples were freeze in liquid nitrogen and stored at -80 oC. Total RNA was isolated from the pericarp after grinding in liquid nitrogen.

  1. 5, r. 174

Reviewer comment:  Incomplete description of how this was accomplished. You describe what you did, but not how.

Gene-specific primers for β-carotene hydroxylase genes were located on chromosomes 3 and 6 of the pepper genome (Supplementary Information 1, Table S1).

Answer to the reviewer:

CrtZchr03 and CrtZchr06 cDNA regions were amplified from total RNA using One-step RT-PCR kit (Qiagen, USA) according to the manufacturer instructions. Briefly, about 1 μg total RNA was mixed with 0.4 μM primers (CrtZ-D_f/CrtZ-D_R for CrtZchr03 gene or BCH1-F/BCH1-R for CrtZchr06 gene). The mix was incubated for 15 min at 95 oC and cooled on ice. 5 μL One-step RT-PCR buffer, 1 μL 10 mM dNTP and 1 μL One-step RT-PCR enzyme mix were added along with ddH20 to a final volume of 25 μL.

Reverse transcription was performed for 25 min at 20 oC and then denatured for 5 min at 94 oC. Then 35 amplification cycles were performed as follows: 1 min at 94 oC, 50 sec at 55 oC, 2 min at 72 oC followed by final extension for 7 min at 72 oC.

The Elongation factor 1 alpha (EF1 - At1g07940) was used as positive control with primers EF1-F: ATT GTG GTC ATT GGY CAY GT and EF1-R: CCA ATC TTG TAV ACA TCC TG (Cho et al., 1995, Kozera and Rapacz, 2013) under same amplification conditions. Amplification products were separated on 1.5 % agarose gel in TAE containing 0.5 μg / mL ethidium bromide. Product length was estimated by comparing with GeneRuler 1 kb DNA Ladder (Thermo Scientific).

  1. 5, r. 178

Reviewer comment:  No description of how cloning and sequencing were accomplished.

Gene cloning and sequencing

Answer to the reviewer:

Amplified fragments were cloned into pTZ57R plasmid vector using InsTAclone kit (Thermo Scientific) and transformed in E. coli DH5alpha. Plasmids from positive clones were isolated and sent for sequencing (StarSeq GmbH, Germany).

  1. 5, r. 181-182

Reviewer comment:  How did you managing to analyze a clone in a mutant genotype?

Clones derived from the variety Oranzheva kapia were analyzed by primer sets CrtZ–D_F/b-CRT 8, CrtZ-E/b-CRT 4 and b-CRT 7/CrtZ-C_R in the mutant genotype of Oranzheva kapia.

Answer to the reviewer:

We have cloned the amplified fragments despite that their size differs from the expected. Sequencing confirmed that these fragments are not related to CrtZchr03 gene. We were not able to amplify fragments from this gene in the mutant while we have successfully amplified them in the initial variety.

  1. 6, r. 225

All local varieties had mature red fruits

  1. 6, r. 227-229

Reviewer comment: Provide the data, with standard errors.

Corrected:

In Material and Methods:

The average mean and the significance interval of the studied characteristics were calculated by SPSS (descriptive statistics). Carotenoid levels were evaluated in individual and group samples [32].

In Results:

When Pazardzhiska kapia 794 and Oranzheva kapia were compared the intervals of average values of the quantified carotenoid substances did not have common point and did not intersect. The obtained data from the confidence intervals of the average values proved that the differences between initial and mutant varieties were statistically significant.

In Discussion:

The results validated the previous study that assessed the mutant gene determined higher β-carotene levels in the mutant genotypes [21]. We evaluated carotenoid levels in individual and group samples [32]. The results validated the previous study that assessed all the genotypes containing the mutant gene that determines higher β-carotene levels.

  1. 6, r. 227-229

Reviewer comment:  According to materials and methods, the run was 20 minutes long. The author only reports ~15 minutes of response. Please explain. The area of the graph beyond 16 minutes is empty space, What concentrations were evaluated? What is mAU?

Answer to the reviewer:

The run time was 15 minutes followed by 5 min. column washing.

  1. 7, r. 256

Reviewer comment:  Table 1. Determination of carotenoid concentration of dry matter (mg / 100 g) of green fruits of Pazardzhishka kapia 794 (parent) and Oranzheva kapia (mutant) varieties (mg / 100 g).

Answer to the reviewer:  Table 1. Carotenoid concentration of dry matter (mg / 100 g) of green fruits of Pazardzhishka kapia 794 (parent) and Oranzheva kapia (mutant) varieties (mg / 100 g).

Reviewer comment:  No replication?  Where are standard errors, etc?

Answer to the reviewer:  Table 1.

added

  1. 8, r. 262

Reviewer comment:  no measure of dispersion

Table 2. Determination of carotenoid concentration in dry matter lyophilized samples from botanical

ripe fruits (mg / 100 g).

  1. 8, r. 267

Reviewer comment:  Figure is fuzzy. What does the height of the red and green bars indicate?

Schematic diagram of analysed carotenoid substances in mutant and parent varieties in ripe

fruits. Partial schematic representation of the carotenoid biosynthetic grid.

Answer to the reviewer: 

Figure 3. Schematic diagram of analysed carotenoid substances in mutant and initial varieties in ripe fruits. Partial schematic representation of the carotenoid biosynthetic grid. Green bars – analysed samples from mutant variety, red bars – analysed samples from initial variety.

  1. 8, r. 269

Reviewer comment:  Figure 3 says nothing about absence of lycopene in the samples. Figure 3 only shows standards, not fruit extracts. I can only conclude that your method is incapable of detecting lycopene.

Answer to the reviewer: 

The figure 3 shows analysed samples. Lycopene was not detected in pepper samples. No information in the bibliography about the detection of lycopene in pepper fruits.

  1. 8, r. 269

Reviewer comment:  Incomplete description of figure.

Figure 5. Amplification patterns of the loci, selected for sequencing.

Answer to the reviewer:  ??? Figure description corrected

  1. 11, r. 347-353

Reviewer comment:  Belongs to Introductory material

The orange-coloured variety Oranzheva kapia was developed via 120 Gy X – ray dose application to Pazardzhishka kapia 794, commonly used local variety in Bulgaria. The mutant plants were back-crossed with parent (3 generations) and self-pollinated for a total of 8 generations, thus generating NIL/RIL [20].

Developed as NIL / RIL, the mutant variety Oranzheva kapia was genetically very close to the parent one Pazardzhishka kapia 794. Their differences are only due to the mutation in Oranzheva kapia.

Answer to the reviewer:  The paragraphs are moved to Introductory section.

  1. 11, r. 357-358

Reviewer comment:  What? I thought we were aimed at paprika production,

Answer to the reviewer: 

The sentence: The fruits are primarily intended for consumption while green, underlines that the pepper is sweet pepper, not paprika for grinding, not red pepper for grinding

Inserted in the beginning of the Introduction is the following: “sweet pepper ‘kapia’ type intended for fresh consumption”, and which would remove any ambiguity. The details are related to Oranzheva kapia and Pazardzhishka kapia 794 varieties – subjects of study.

It is also added about Kurtovska kapia 1 - It also belongs to ‘kapia’ type a sweet pepper.

The capsanthin content is of importance for the paprika for grinding. Here we work with sweet pepper for fresh consumption, that may be also processed.

  1. 11-12, r. 362-363

Reviewer comment:  Reference

Answer to the reviewer:  added [37]

Added ‘carrot’

  1. 12, r. 368

Reviewer comment:  Do you mean pigment composition?

The changes of the pigment spectrum were

Corrected: 

The changes of the pigment composition were

  1. 12, r. 371

Reviewer comment:  Not without statistics.

Significantly

Corrected:  Statistics added

  1. 12, r. 377

Reviewer comment:  Really, you have confidence in 4 decimal places?

Corrected:  They are decreased to three decimal places in the manuscript.

  1. 12, r. 379

Reviewer comment:  NO, it is similar to other lines carrying the orange phenotype and mutation present in O k.

This is similar to the results of mutant lines originating from crosses of Oranzheva kapia

Answer to the reviewer:  Not every orange phenotype will be with high beta-carotene. In some cases the red colour can be less intensive (the carotenoids with red colour will contribute less) and together with the yellow colour it will result into orange frui colour. In this case the beta-carotene (the orange pigment) is not increased and it is not the reason that determines the orange fruit. An eventual mutation in CCS may give this effect.

  1. 12, r. 393

Reviewer comment:  Should have been presented in introduction.

Corrected:  If this statement is in introduction the following will not be well understood:

The mechanisms of the observed 7.5 fold increase of α-carotene in the mutant compared to the parent are not well understood.

  1. 13, r. 410

Reviewer comment:  What are parent mutants?

Answer to the reviewer:  parent is replaced in the manuscript into initial mutants

parent mutants removed and ‘initial fruits’ used.

  1. 13, r. 421

Reviewer comment:  Why?

The mutation is likely to affect the presence of the enzyme or the regulation of .-carotene

hydroxylase enzyme activity.

Answer to the reviewer:    These two cases are the only logical. I don’t know other opportunity that results in a similar phenotype.

  1. 13, r. 421

Reviewer comment:  early stages

Answer to the reviewer:    ‘early stages of plant development’ added to the existing text

  1. 13, r. 431

Reviewer comment:  Probably should not have concluded earlier that it was a deletion. What kinds of mutations are expected with X-ray?

Sentence: is an event leading to the deletion, inversion, translocation or insertion of a retrotransposon into the sequence of the target gene of the mutant, occurring frequently after physical mutagen treatment.

Answer to the reviewer:  After applying Rӧ-rays irradiation the mutations are frequently deletions of about 50 bp fragment(an old statistics). However, every type of mutation can occur after X-rays treatment. The mutation cannot be predicted. can occur

  1. 13, r. 433

Reviewer comment:  Relevance of this statement

This study’s results were obtained after amplification, cloning and sequencing of the two .-carotene hydroxylase genes.

Answer to the reviewer:  Not relevant. Removed.

  1. 13, r. 440

Reviewer comment:  What is this, explain?

Answer to the reviewer:    A709G transition. - this is the exact nucleotide position along the length of the target gene affected by the mutation that is a transition - A replaced by G at 709 nd position. This sign explains what nucleotide is replaced by what as a mutation event.

  1. 14, r. 451

Reviewer comment:  Seems to me this paper should have been evident in the introduction.

Removed sentence: The lack of amplification at several points within the gene eliminates translocation and inversion as possible reasons.

Corrected sentence: Lack of amplification at several regions within CrtZchr03 locus suggests gene deletion instead of translocation or inversion.

Answer to the reviewer:    This means that our Results and Discussion are very logical and well understood. We did not know this result all these before the amplification, cloning and sequencing. There are so many different options that could be, instead of the ones found here.

Probably should have indicated earlier why this line of research was pursued.

The lack of amplification at several points within the gene eliminates translocation and inversion as possible reasons. Insertion of a mobile element or another fragment is also eliminated.

  1. 14, r. 457-458

Modern breeding programs need detailed characterizations of available germplasm. This

includes not only genotyping and markers but also characterization of mutations for better

understanding of processes in the plant and expected output.

I agree. And any phenotypic characterization should include estimates of dispersion.

  1. 5, r. 461

One way to meet breeder’s need is to carefully

design experiments and to select proper tools from the enormous molecular biology arsenal.

  1. 5, r. 463

Corrected Conclusions:

The application of HPLC revealed that the orange fruit color in the mutant variety Oranzheva kapia correlates with high α- and β-carotene levels that are several times higher than in the parent fruits. While the level of β-carotene increased dramatically, the concentration of three xanthophylls (lutein, β-cryptoxanthin, and zeaxanthin) decreased in the mutant fruit. Thus, consuming two to three fruits of the mutant pepper should meet the daily β-carotene requirement.

The performed PCR analysis allowed us to propose that the X-ray mutagenesis of parent variety Pazardzhiska kapia 794 resulted in deletion of the CrtZch03 gene in the mutant variety Oranzheva kapia along with neighboring regions on chromosome 3. This deletion abolished the carotene biosynthetic pathway that led to the accumulation of α- and β-carotene and the resultant orange color of the mutant fruits. Henceforth, induced mutagenesis still remains an important tool for enhancing nutritional quality in the peppers.

Reviewer 2 Report

This was an interesting paper examining the concentrations of b-carotene in a mutant plant line identifying a critical gene within the b-carotene pathway.

Please find attached PDF with comments.

Author Response

Replies to Review 2

  1. 2, r. 70

Reviewer comment: is this supposed to be 'during'

Answer to reviewer: yes, corrected

  1. 3, r. 85

Reviewer comment:  Two NIL/RIL...

Answer to reviewer:  yes, corrected

  1. 3, r. 94-95

Thank for the corrections!

  1. 3, r. 101

Corrections (2) -done.

  1. 4, r. 141

Correction -done.

  1. 5, r. 159

Suggested by the reviewer and the correction done: 

In order to determine the percentage of recovery samples were spiked with each carotenoid prior to the extraction procedure.

  1. 5, r. 164

Suggested by the reviewer and the correction done: 

gDNA, cDNA and RNA isolation.

Include details of how cDNA was produced.

Recommended information included.

2.3.1. DNA and RNA isolation.

Genomic DNA was isolated from freeze-dried young leaves according to [26, 27]. Lambda DNA (Thermo Fisher Scientific, Lithuania) was used to determine for DNA quantity on 1 % LE agarose gels (Lonza, USA) with ethidium bromide (VWR International, Austria). DNA concentration in the samples was estimated by comparing band intensity of probes with lambda DNA of known concentrations (50 ng / μL, 20 ng / μL, 10 ng / μL and 5 ng / μL). 3 μL of each dilution were loaded per lane.

Fresh samples were frozen in liquid nitrogen and stored at -80 oC. Total RNA was isolated from the pericarp after grinding in liquid nitrogen. RNA from a 50 mg sample was purified with a by Qiagen RNase Plant Mini Kit. RNA concentration was determined spectrophotometrically at 260 nm.

  1. 5, r. 180

Suggested by the reviewer and the correction done:  S7

  1. 5, r. 197-201

Exact PCR conditions should be provided for each primer pair so that other researchers can reproduce this work.  It can be supplied as a supplementary table.

Suggested by the reviewer and the correction done: provided in Supplementary file – S7-8 Supplementary Materials 2 - Primers - added Tm. Tm of the primers added that will enable conducting the reactions by another researcher.

  1. 6, r. 216

Reviewer comment:  Details of the kit.

Recommended information included: We have cloned the amplified fragments despite that their size differs from the expected. Sequencing confirmed that these fragments are not related to CrtZchr03 gene. We were not able to amplify fragments from this gene in the mutant while we have successfully amplified them in the initial variety.

  1. 6 r. 225-226

Suggested rewording:  accepted.

  1. 7 r. 234

Reviewer comment:  Neither image is labeled.  It looks like Pazardzhishka in figure a has both red and orange fruit.  Can you please explain this?

Corrections done: Figure name corrected:

Figure 1. Mature fruit morphology of the two pepper varieties – (a) a red fruit characteristic for the variety Pazardzhishka kapia 794 (I) and an orange fruit of Oranzheva kapia (M), (b) - orange-fruited Oranzheva kapia (M) (20 fruits were collected and numbered for the HPLC analysis).

what Figure number?

Figure 2

  1. 7, r. 244

Figure 3 – the number corrected

  1. 7, r.256

Reviewer comment:  Is this the average lutein and carotene concentrations or from a single sample?  If an average then indicate n = ?, and standard error and do a T-test to indicate if they are statistically different between varieties.

Corrections done: Table 1 – details included

A column removed

Formatting improved

  1. 8, r. 261

Reviewer comment:  As with table 1.  Is this an average and provide further information.

Corrections done: information about variation included.

  1. 8, r. 267

Corrections done: Figure 3 – quality improved

  1. 9, r. 276-278

Recommended corrections done:

  1. 9, r. 280-282

Recommended corrections done:

Figure 4.

  1. 9, r. 283

Please provide details of which primers were used in each set of reactions.  It may also be useful to have a schematic of were the primers lie on the Pazardzhishka sequence.

A new table added for the requested additional information about the PCR reactions.

  1. 9, r. 285

Recommended corrections done:

  1. 10, r. 315

This was not included in the supplementary files downloaded.

Recommended corrections done:

  1. 11, r. 333

Recommended corrections done:

  1. 11, r. 334

Are the negatives indicating lane spaces between samples or no-template controls?

No amplifications in the mutant

  1. 11, r. 336 and 349, 355-356

Recommended corrections done:

Round 2

Reviewer 1 Report

Please indicate number of biological replications for pigment analysis and for molecular work. 

One sentence paragraphs are unacceptable. 

The discussion is not focused, is repetitive and can be reduced by at least 50%

It appears dispersion values are not based  on experimental error, but on analytical error.

Author Response

First review

Replies of authors

Corrections done

Thank you for your efforts and the corrections helping us to improve the manuscript!

Reviewer - One sentence paragraph; 287 r., 299 r., 313 r., 316 r., 321 r., etc. -  Another single sentence paragraph.

Authors – All ‘One-sentence paragraphs’ were corrected in the text.

Authors – Additional text explaining the figure 1 added.

303 r. Reviewer - Still an incomplete explanation. What does the height of the bars indicate?

Authors - Additional text explaining the figure 2 added.

Authors - Quantities evaluated added in the text.

309 r. Reviewer - You can not state that it was absent. It was below your LOD.

Authors - Additional text for lycopene, 0.010 μg / mL added to note in the text that it was below our LOD.

Table 1 and Table 3 – corrected; data added; confidence interval, then shown; significant digits reduced;

The tables show mean values and their interval of confidence at p <0.05. This is not a standard deviation.

334 r. corrected – progenitor removed – initial added (as initial is a term used in induced mutagenesis.

337 r. – corrected letter ‘w’ – with ‘v’ – visualization

359 r. – sentence corrected

The first paragraph in the beginning of section 3.2. Characterization of the mutation, leading to orange-coloured fruits’ was corrected.

377 r. Reviewer - Sentence makes no sense. Authors - corrected.

Reviewer - indicated is too strong. Perhaps "consistent with"?

Authors – corrected sentence. Our previous study [36] stated the recessive inheritance of the mutant allele.

420 r. Reviewer - They cannot be primarily consumed as green and preferred as ripe at the same time. Authors – corrected sentence.

440 r. corrected sentence.

471 r. corrected sentence.

476 r. Reviewer - What do the pink and green boxes represent? Authors – corrected text added.

479 r. Reviewer – What is the relevance of this study? Authors – corrected - text removed.

498 r. corrected - text removed.

500 r. – text corrected.

510 r. – text corrected.

539 r. – corrected – text removed.

Correction done in section References - https://bioversityinternational.org/fileadmin/_migrated/uploads/tx_news/Descriptors_for_capsicum__Capsicum_spp.__345.pdf

Reviewer 2 Report

Thank you for addressing the suggested corrections.  The manuscript is greatly improved.  Below are a few further minor corrections:

Line 534.  This sentence does not match the results in the table.

Line 733 and 738. Correct spelling of visualisation.

Line 820. Do you mean 'standard'?

Line 868-869.  There appears to be a repetition of words.

Line 921.  Rearrange this section so that the concentration in the peppers and carrots are together for comparison. 

Line 935.  remove the second 'the' so that it reads '...the only gene...'

Line 1058. suggesnitialtion?  Do you mean hypothesis?

Author Response

Second review

Replies of authors

Corrections done

Thank you for your careful reading, pear review and help to improve further the manuscript!

Line 534. Reviewer: This sentence does not match the results in the table. Authors: Corrected.

Line 733 and 738. Reviewer: Correct spelling of visualisation. Authors: Corrected.

Line 820. Reviewer: Do you mean 'standard'? Authors: Yes, corrected.

Line 868-869. Reviewer: There appears to be a repetition of words. Authors: Corrected repetitions of words in the text (line 868-859 was not found).

Line 423. Reviewer: Rearrange this section so that the concentration in the peppers and carrots are together for comparison. Authors: Text was rearranged and we believe in a better way.

Line 935. Reviewer:  remove the second 'the' so that it reads '...the only gene...' Authors: It was removed.

Line 520. Reviewer: suggesnitialtion? Do you mean hypothesis? Authors: This was corrected. Correct words - suggested an inactivation

Round 3

Reviewer 1 Report

There is still no indication of biological replications actually used for molecular work or for HPLC analysis. I would not permit publication until these are explicitly provided. Single sentence paragraphs still remain, but some have been eliminated. I have marked a few items in the manuscript, mainly for the editor to correct. English could be improved, and I will leave that to the editor.
